# *Defects4C*: Benchmarking C/C++ Faults to Assess LLM-Based Program Repair

## Abstract

Automated Program Repair (APR) plays a pivotal role in ensuring the quality and reliability of software. However, most existing APR research focuses on Java programs, primarily due to the well-established benchmark such as Defects4J. Despite the significant prevalence of C/C++ vulnerabilities, the field lacks extensive research on the automated repair of such vulnerabilities, primarily attributed to the absence of high-quality open-source benchmarks in this domain.

To address the critical gap in available datasets for C/C++ program repair, this paper introduces *Defects4C*, a comprehensive and high-quality executable benchmark designed to improve defect detection and repair. The dataset includes a vast collection of bug-relevant commits (e.g., **9M** in total), **248** high-quality buggy functions and **102** vulnerable functions paired with test cases for reproduction. These datasets can be used to evaluate repair techniques and to retrain learning-based methods for improved performance. Using this expanded dataset, we evaluate the performance of state-of-the-art LLM-based automated program repair techniques in addressing C/C++ faults. Specifically, we conduct an extensive empirical study with **24** leading LLMs. Our findings provide valuable insights into the capabilities and limitations of existing APR approaches for C/C++ programs, underscoring the necessity for novel APR techniques and the significance of *Defects4C*. This dataset marks a significant advancement in the field, offering a robust and comprehensive C/C++ dataset that is instrumental for future research on program repair.

## 1 Introduction

Software bugs pose potential security threats to software systems. Automating the detection and repair of software bugs is crucial in software development and has attracted widespread attention from academia and industry. Many repair works powered by various techniques have been proposed (Just et al., 2014; Tufano et al., 2019), targeting to accurately and efficiently repair bugs in programs to increase software developer productivity and reduce the debugging costs. Moreover, the advent of large language models (LLMs) has demonstrated significant improvements over traditional repair methods, offering superior performance in program repair tasks (Xia & Zhang, 2024).

To evaluate the effectiveness of the proposed automated program repair (APR) techniques, some benchmarks in different programming languages are constructed and released (Tufano et al., 2019; program repair.org, 2021) for users to evaluate. For instance, Defects4J (Just et al., 2014) has confirmed its dominance as a standard benchmark where the majority of repair-related techniques in Java programming language utilized it for comparison (An et al., 2023). BugsInpy (Widyasari et al., 2020) is another collection of defects from real-world Python projects to evaluate the repair performance in Python programming language.

It is noteworthy that, according to the report (mend, 2024), C is the language that has the most reported vulnerabilities among all, accounting for more than 50% of all reported open source vulnerabilities since 2019. Furthermore, the annual count of vulnerabilities in C significantly exceeds that of any other programming language. Given the significant threat posed by vulnerabilities in the C language to software systems, some efforts have been made to construct C / C++ defect benchmarks to evaluate existing APR techniques (Orvalho et al., 2022; Tan et al., 2017; Böhme et al., 2017; Yi et al., 2017; Le Goues et al., 2015; An et al., 2023; Gupta et al., 2017). However, challenges remain. Notably, some of these benchmarks, such as DeepFix (Gupta et al., 2017) and Code4Bench (Majd et al., 2019),

source their bugs from student assignments or competitive programming platforms like Codeforces, resulting in simpler buggy functions that lack the complexity of real-world applications. Several benchmarks based on real-world projects have been introduced (Böhme et al., 2017; Le Goues et al., 2015; An et al., 2023; Long & Rinard, 2016), but they still have limitations. For instance, DBGBench (Böhme et al., 2017) collects data from only two projects, leading to incomplete and insufficient evaluation across diverse software ecosystems. ManyBugs (Le Goues et al., 2015) and Prophet (Long & Rinard, 2016) include C/C++ programs in limited versions (e.g., only C99 and C11 in ManyBugs) and have limited usability (e.g., requiring long compilation for every patch test and lacking a user-friendly command-line interface), which complicates test validation and usage (Lutellier et al., 2020). The latest benchmark, BUG-C++ (An et al., 2023), sources defect data from GitHub commits but lacks human verification to confirm whether the identified issues are actual bugs. Our preliminary studies reveal that some of these changes are unrelated to bugs and instead involve functionality updates. In summary, there remains a pressing need for a high-quality C/C++ fault benchmark that meets the criteria of practicality, diversity, fidelity, and usability.

Automated program repair techniques, designed to automatically resolve software bugs, have evolved significantly with the rise of large language models such as ChatGPT (OpenAI, 2022). Studies on code understanding and generation highlight the remarkable capabilities of these LLMs in these areas (Chen & Zaremba, 2021; Liu et al., 2023; Xia & Zhang, 2024). Recent research suggests that LLM-based APR techniques outperform traditional approaches in both bug-fixing efficiency and accuracy (program repair.org, 2021). However, most of these techniques are evaluated using Defects4J (Just et al., 2014), which is favored for its collection of high-quality bugs (357 in Defects4J 1.0) and its user-friendly command-line interface that facilitates quick and convenient assessment of model-generated repairs. Despite these advances, the lack of a similarly high-quality dataset for C/C++ has left the effectiveness of LLM-based APR techniques in C/C++ programming largely under-explored. This gap prevents researchers from fully understanding the capabilities of LLMs and challenges in C/C++ program repair. Given the large number of bugs in C/C++ programs and their unique characteristics, it is crucial to evaluate these techniques on C/C++ faults to fully uncover their potential and drive further advancements in the field.

To address the identified challenges, we introduce a new high-quality C/C++ fault benchmark, referred to as *Defects4C*, which consists of two major components: bug-relevant commits (*Defects4C_bgcommit*), and high-quality buggy functions that are further divided into general bug functions (*Defects4C_bug*) and vulnerability functions (*Defects4C_vul*). Specifically, the commits dataset *Defects4C_bgcommit* may include some false positives, making it suitable for model training or fine-tuning, while the buggy functions (i.e., *Defects4C_bug* and *Defects4C_vul*) are rigorously confirmed by human experts, ensuring their reliability for strict evaluation purposes.

Specifcially, we leveraged BigQuery to extract a large number of buggy commits (*40M*) from over *110K* widely used GitHub C/C++ repositories using a set of predefined bug-related keywords. We then filtered the commits based on availability (resulting in **9M** bug-related commits) and whether the changes were isolated to a single function (leading to **76K** single-function buggy commits). A unit test matching method was applied to identify corresponding test cases for each buggy function, leaving representative **3,785** buggy commits collected from the top 100 projects with paired tests. To ensure the quality of the dataset for evaluation, we implemented a three-stage human annotation process conducted by three security experts. This process was crucial for eliminating false positives, i.e., cases where commit messages contain bug-related keywords, but the code changes do not actually address bugs or security issues. Our rigorous approach resulted in **248** confirmed bugs (*Defects4C_bug*) along with their corresponding unit tests, allowing for bug reproduction and validation.

In addition, we expanded the diversity of the dataset by including a vulnerability dataset (*Defects4C_vul*). We first extracted C/C++-related Common Vulnerabilities and Exposures (CVEs) from a publicly available database (CVEProject, 2021). To isolate vulnerable functions, we selected CVEs that provided patched commit IDs, allowing us to retrieve the associated vulnerable and patched functions from the commits. We then applied the unit test matching process to identify corresponding test cases for each vulnerability, ultimately yielding **102** vulnerabilities with corresponding unit tests.

To understand the effectiveness of state-of-the-art LLM-based APR techniques in fixing C/C++ bugs or vulnerabilities, we conducted an empirical study using our *Defects4C* benchmark. The study focuses on evaluating the performance of LLM-based APR techniques, incorporating **24** state-of-the-art LLMs. These models are evaluated in single-round and conversation-based program repair

Table 1: Existing C/C++ benchmarks for program repair.

| Dataset | Defects | Projects | Source | Dataset | Defects | Projects | Source |
|---|---|---|---|---|---|---|---|
| CodeHunt (Tillmann et al., 2014) | 195K | N/A | Interview/Contest | ITSP (Sykes & Franek, 2003) | 661 | N/A | Assignment |
| Code4Bench (Majd et al., 2019) | 25K | N/A | Interview/Contest | C-Pack-IPAs (Orvalho et al., 2022) | 513 | N/A | Assignment |
| Prutor/SARD (Das et al., 2016) | 23K | N/A | Interview/Contest | Bugs-C++ (An et al., 2023) | 209 | 22 | Real-World |
| SPoC (Kulal et al., 2019) | 18K | N/A | Interview/Contest | ManyBugs (Le Goues et al., 2015) | 185 | 9 | Real-World |
| CodeFlaws (Tan et al., 2017) | 3.9K | N/A | Interview/Contest | Prophet (Long & Rinard, 2016) | 69 | 8 | Real-World |
| DeepFix (Gupta et al., 2017) | 6.9K | N/A | Assignment | DBGBench (Böhme et al., 2017) | 27 | 2 | Real-World |
| IntroClass (Le Goues et al., 2015) | 998 | N/A | Assignment | *Defects4C* | **350** | **41** | Real-World |

scenarios with various experimental settings. Our findings reveal a significant performance gap in LLM-based APRs when addressing C/C++ faults compared to their success with the Defects4J benchmark (Java). This discrepancy highlights the urgent need for APR techniques specifically tailored for C/C++ fault repair. We further explored the effectiveness of fine-tuning in C/C++ program repair, and while the results show some promise, they remain below acceptable levels. Our newly developed *Defects4C*, with its high-quality and comprehensive dataset, is positioned to serve as a valuable resource for future research in C/C++ program repair.

To sum up, we make the following contributions:

- We have developed and publicly released an **executable** C/C++ defect benchmark namely *Defects4C*, comprising **9M** bug-relevant commits (*Defects4C_bgcommit*), **248** buggy functions (*Defects4C_bug*) and **102** vulnerable functions (*Defects4C_vul*), sourced from GitHub open-source projects. It is accessible at the website[1]. A user-friendly command line interface for ease of use accompanies each sample in this dataset.

- We conduct the first large-scale empirical study focused on assessing the capability of LLM-based APR techniques in repairing C/C++ programs. We select **24** state-of-the-art LLMs with various settings for a comprehensive evaluation. Our findings highlight a significant gap and limitations in the current LLMs when fixing C/C++ bugs, especially in contrast to their performance on Java bugs. These results underscore the urgent need for further research and development of C/C++-specific repair techniques and the importance of our benchmark.

## 2 RELATED WORK

**Existing C/C++ Defect Benchmark.** Table 1 provides a summary of existing C/C++ benchmarks for program repair, including our proposed dataset, *Defects4C*. To date, prevailing benchmarks for C/C++ programs have mostly centred on student programming assignments such as DeepFix (Gupta et al., 2017), C-Pack-IPAs (Orvalho et al., 2022), IntroClass (Le Goues et al., 2015) or online contests such as Code4Bench (Majd et al., 2019), CodeHunt (Tillmann et al., 2014), Prutor/SARD (Das et al., 2016), SPoC (Kulal et al., 2019), CodeFlaw (Tan et al., 2017). As the data source is from assignments or contests, they are impractical with relatively low practical value in real-world program repair. To construct a more practical benchmark, several works propose to construct it from real-world projects such as ManyBugs (Le Goues et al., 2015), Prophet (Long & Rinard, 2016), DBGBench (Böhme et al., 2017) and BUG-C++ (An et al., 2023). These benchmarks also suffer from various limitations. For instance, ManyBugs and Prophet offer low usability and only support outdated versions of C/C++ programs. DBGBench is limited in diversity, as it is collected from only two GitHub projects. BUG-C++ lacks rigorous verification, as it mainly relies on bug-related keywords from commit messages without confirming whether the collected issues are actual bugs.

**LLM-based Program Repair.** Large language models (i.e., LLMs) have exhibited powerful capabilities to repair program defects (Jiang et al., 2023; Prenner et al., 2022; Sobania et al., 2023; Xia et al., 2023). Compared with single-round repair, recent conversation-based program repair techniques (Xia & Zhang, 2023; 2024) are proposed to improve the repair performance further. These techniques target interaction with LLMs by feeding error messages as the input to guide LLMs in generating more accurate output. Although various LLM-based techniques are proposed for program repair, they are mainly based on Defects4J (Just et al., 2014), a widely used defect benchmark for Java programs.

---

[1]https://sites.google.com/view/anonymous-defects4c

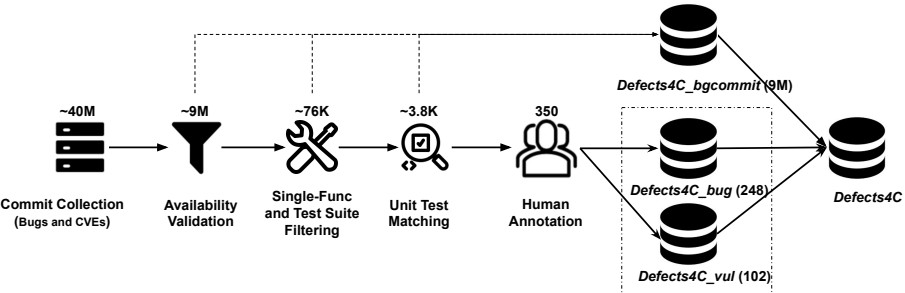

Figure 1: The pipeline of data collection process.

## 3 BENCHMARK CONSTRUCTION

### 3.1 RAW DATA COLLECTION

**Commit Collection (40M)** To extract buggy functions, we follow previous works (Zhou et al., 2021; An et al., 2023) and collect GitHub commits related to bugs. We primarily use BigQuery to extract commits from two types of projects: open-source, non-fork C/C++ repositories with redistributable licenses, having 200+ stars, from January 2015 to August 2023 (sourced from the GH Archive (GH Archive, 2023)), and the top 500 C/C++ projects ranked by GitHub stars (EvanLi, 2016). The 500 high-ranking projects were included to ensure that BigQuery did not miss such popular projects. Based on the selected projects, we applied a keyword-based heuristic rule inspired by VRepair (Chen et al., 2022) to filter out commits unrelated to bugs. We considered commits as plausible bug-related if their messages contained keywords such as *fix*, *solve*, *repair*, *bug*, *issue*, *problem*, *error*, *fault* and *vulnerability*. Using this method, we obtained **38M+** commits from these projects, with a total cost of approximately *$5,000* to gather the data via BigQuery.

To construct the vulnerability dataset, we selected CVEs related to C/C++ programming from the CVEProject repository [2], which includes CVEs collected from 1999 to 2024. We only selected CVEs that provided a single patched commit ID, resulting in a total of **14,488** commits. This choice was made for two reasons: first, the CVEs with a commit ID allow us to retrieve the specific vulnerable functions, and second, if a CVE had multiple commit IDs, it would be hard to confirm which commit ID was used to address the vulnerability. Finally, we collected about **40M** raw commits that are related to bugs.

**Commit Validation (9M)** We recognize that the commits obtained through BigQuery or the CVE website may become invalid over time due to factors such as repository ownership transfer, archiving, or other reasons. Therefore, we filter out these invalid commits based on their availability. Additionally, we implement a rigorous deduplication process to remove duplicate commits, resulting in a total of **9M** valid bug-relevant commits. From these commits, we can extract function pairs from before and after the commits, which represent the buggy and patched versions, respectively. Although some false positives remain due to the keyword-based filtering process, these commits are still valuable for fine-tuning APR models, especially there is no an existing real-world bug repair dataset for retraining. However, they are not suitable for evaluation due to the lack of rigorous bug verification.

**Single Function Commit Filtering (76K)** The collected commits may involve modifications across multiple files, which are too complex for existing APR techniques. To reduce complexity, we retain only those commits that involve changes to a single function. To ensure the extracted functions are executable and verifiable, we further filter out commits that lack an associated test suite for validation. Through this process, we identify **76K** valid commits, including **249** vulnerability-relevant commits.

### 3.2 UNIT TEST MATCHING

Each commit is associated with a test suite containing multiple test cases, as established through the commit validation and filtering process in Section 3.1. However, identifying which specific test case verifies the current fix is necessary, as many test cases are designed to validate functionality changes across the project's entire history. While some straightforward identification methods exist, such as in Java projects, where a function named *abc* is often tested by a test case named *test_abc*, this

---

[2]https://github.com/CVEProject/cvelist

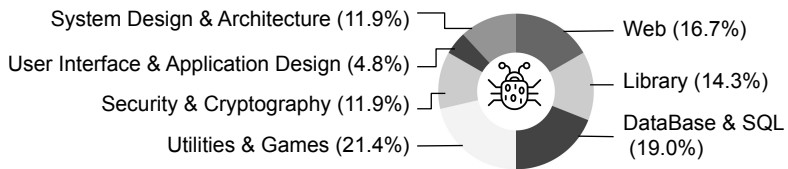

Figure 2: The category constitution of our *Defects4C*.

naming convention is not commonly found in C/C++ projects, making it impractical to apply. Hence, we propose an unit test pair verification algorithm, which is based on a basic observation: *for a buggy/vulnerable fix, there typically exists some unit tests that pass on the corrected code version but fails on the buggy/vulnerable version.* Specifically, for a given test suite $T = \{t_1, t_2, ..., t_n\}$, where $t_i$ is the test case, a commit yielding two code versions: $V_0$ (pre-commit) and $V_1$ (post-commit), representing the previous and post version of code after the commit, respectively. For each test case in $T$, if $t_i$ passes $V_1$ but fails to pass $V_0$, we consider it the test case used to evaluate current fixing. We consider the other cases as bug-unrelated test cases and filter them out. Finally, we can obtain a subset $T'$ from $T$ where each test case is used to evaluate the buggy function. Following this rigorous process, we further validate the *76K* data obtained from Section 3.1 and get **3,785** commits for *Defects4C_bug* and **102** commits for *Defects4C_vul*.

### 3.3 HUMAN ANNOTATION

Given the potential for false positives in the bug-related commits, we conducted a conservative and rigorous human annotation process to reproduce, confirm and classify the bugs, ensuring a high-quality evaluation dataset for APR techniques. Specifically, following prior studies (Quan et al., 2022; Shi et al., 2022), we divided the collected commits from Section 3.2 (3,785 and 102) into two equal parts and applied a three-round annotation process. In the first round, half of the dataset was assigned to two security experts for independent confirmation and classification of bugs, following the CWE bug types, with a focus on the root cause and fixed logic. The experts then discussed their classifications and determined which bugs should be included in the dataset, with any disagreements resolved by an arbitrator. To assess the consistency of their classifications, we used Cohen's Kappa (k) coefficient (Hsu & Field, 2003), which measures inter-rater reliability, where higher values indicate greater agreement.

In the first round, the inter-rater reliability (k) was 0.48. After establishing a preliminary taxonomy, the experts manually annotated the remaining half of the dataset in the second round, improving the k coefficient to 0.60. In the third round, the experts performed a resampling exercise, reviewing 50% of the reported bugs twice to verify the results, achieving a k value of 0.88, which indicates almost perfect agreement (Landis & Koch, 1977).

Through this human annotation process, we identified several issues with the commits. Some commits, despite containing bug-related keywords, only added features or modified output formats without fixing actual bugs. Others had vague commit messages (e.g., "fix bug") that did not logically align with the code changes, or were reverted in later iterations, making them unreliable bug fixes. After completing the annotation process, we identified **248** commits for *Defects4C_bug* and **102** commits for *Defects4C_vul*. Notably, we did not filter any vulnerability commits, as they were sourced from the high-quality CVE repository. In total, we obtained **350** high-quality faults that are reproducible and suitable for evaluation by APR techniques.

## 4 STATISTICS OF *Defects4C*

Finally, *Defects4C* comprises **9M** bug-related commits for *Defects4C_bgcommit*, **248** commits for *Defects4C_bug*, and **102** commits for *Defects4C_vul*. Among these, *Defects4C_bgcommit* includes **76K** single-function commits with potential test suites, and **3.8K** commits with executable tests.

We also conducted a statistical analysis of the evaluation data, specifically *Defects4C_bug* and *Defects4C_vul*. Firstly, we manually classify the application categories of the data, which is presented in Figure 2. We can find that the error code is from diverse application scenarios. We further analyse the types of errors in these data, presented in Table 2, enumerating the specific taxonomy and statistical summary within each category. For a software bug fix, the location where the code has been modified often correlates with the root cause that triggers the bug (Hirsch & Hofer, 2020;

Mahbub et al., 2023). Therefore, we first categorize the dataset into four primary categories based on the logical location of the code changes during the bug fixes, which are defined as `Signature`, `Sanitizer`, `Memory Error`, and `Logic Organization`, respectively. For each primary category, we further divide the classifications into subcategories and align them closely with the root causes of the bugs, where the CWEs served as the standard reference for the detailed taxonomy. Please refer to Appendix A.2 for more introduction to these categories.

Furthermore, in line with prior works (Xia et al., 2023; Xia & Zhang, 2024), we also classify the characteristics of the data in *Defects4C* from the perspective of their code fix locations into three categories, namely *Line*, *Hunk*, and *Function*. Specifically, *Line* represents the bugs where the fixing code is completed within a single line, *Hunk* denotes the fixes with multi-lines and continuous code modifications, and *Function* shows the fixes involve multiple modifications at several places within a single function. We further provide the error distribution across different C/C++

Table 2: The number of bugs and vulnerabilities for different categories.

| Category | Error Type | Bugs | Vulnerabilities |
|---|---|---|---|
| Signature | Incorrect Function Usage | 19 | 3 |
| | Fault Input Type | 12 | 2 |
| | Incorrect Function Return Value | 19 | 3 |
| | Incorrect Variable Usage | 25 | 3 |
| Sanitizer | Control Expression Error | 66 | 6 |
| Memory Error | Null Pointer Dereference | 6 | 6 |
| | Uncontrolled Resource Consumption | 9 | 5 |
| | Memory Overflow | 5 | 61 |
| Logic Organization | Improper Condition Organization | 67 | 11 |
| | Wrong Function Call Sequence | 20 | 2 |

projects in Appendix A.1. Lastly, we conduct a lot of engineering work to make a user-friendly command line interface (i.e., CLI) to ensure each bug and vulnerability can be reproduced easily. We provide the details in Appendix A.3.

## 5 EVALUATION SETUP

Large language models have demonstrated significant effectiveness in APR (Xia et al., 2023; Xia & Zhang, 2023; 2024). Therefore, we further evaluate their performance on C/C++ repair tasks using our *Defects4C* dataset. In particular, we first assess the performance of existing methods that rely on pre-trained LLMs using our evaluation datasets *Defects4C_bug* and *Defects4C_vul*. Additionally, we fine-tune the LLMs using *Defects4C_bgcommit* to explore whether this improves their performance.

### 5.1 SETTINGS FOR PRE-TRAINED MODELS

In this setting, we directly utilize LLMs without fine-tuning for assessment. In particular, we select a large number of state-of-the-art LLMs (**24**) to evaluate their performance. The assessment is categorized into single-round and conversation-based repair.

Single-round repair refers to the model generating a patched program once based on the given prompt, without receiving feedback or undergoing multiple iterations of verification and re-generation. Similar to EvalPlus (Liu et al., 2023), we use the unbiased pass@$k$ (Chen & Zaremba, 2021) to assess the LLM-synthesised code's repair performance accurately. We conduct random sampling to generate 100 program repairs for each of two temperature settings (0.2, 0.8) and greedy-search decoding. For random sampling, we present the best-performing pass@$k$ for each $k \in \{1, 10, 100\}$ and its corresponding temperature denoted by $T_k^*$. For greedy decoding, which only generates one output, we evaluate its pass rate as pass@$k^* = 1$. GPT-4 is only evaluated under greedy decoding due to the time and cost constraints.

Conversation-based repair, as described by Xia et al. (Xia & Zhang, 2024), involves invoking the model multiple times. In each iteration, the error feedback from the compiler in the previous round is incorporated into the prompt provided to the LLMs, helping to generate more accurate outputs in the current round. It is costly to use pass@$k$ as the evaluation metric in this setting because pass@$k$ requires generating a massive amount of model outputs. Hence, we follow Xia et al. (Xia & Zhang, 2024) to report the number of successful repairs in *Defects4C*. Specifically, we select the best-performing models from different model series within the single-round repair used to evaluate the conversation-based repair due to the cost. The temperature is set to 1.0 following the configuration (Xia & Zhang, 2024). We add another greedy decoding strategy to evaluate the effect of different decoding strategies in the conversation-based repair. Our default setting for the maximum number of repair attempts is 10, and the maximum conversation length in each attempt is

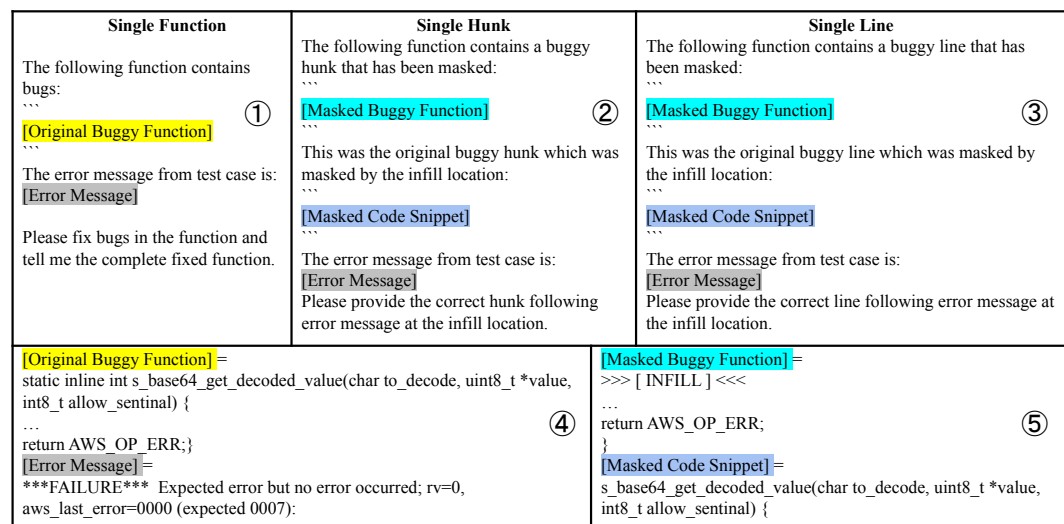

Figure 3: Prompt design for different types of defects.

3. Consequently, we conduct 30 repair attempts for each buggy function until an output that passes all test cases is generated. For more details about the conversation repair, please refer to Appendix B.

The experiments are conducted on a server with 8 RTX A6000 GPUs. The batch size is 16, and the maximum input sequence length is 2048 for all experiments. Please refer to Appendix A.3 for more experimental configurations.

## 5.2 SETTINGS FOR FINE-TUNING

The majority of LLM-based APR research relies on pre-trained models, primarily due to the lack of datasets capable of supporting large-scale fine-tuning for repair tasks. However, our dataset *Defects4C_bgcommit* addresses this limitation. Therefore, we further conducted a study to evaluate repair performance with fine-tuning. Specifically, we selected single-function commits paired with test suites from *Defects4C_bgcommit* as the fine-tuning dataset and evaluated the performance of the fine-tuned models on *Defects4C_bug* and *Defects4C_vul*. Following the approach used in Magicoder (Wei et al., 2023), we performed decontamination to exclude any samples that are identical to, or share similar buggy or patched code snippets with, those in *Defects4C_bug* and *Defects4C_vul* to prevent data leakage. This was achieved by employing UniXcoder (Guo et al., 2022) to embed code snippets and filtering out samples with a cosine similarity score higher than 0.95 when compared to samples in *Defects4C_bug* and *Defects4C_vul*. In addition, after filtering the input length greater than 2048, we retained 20,591 samples from *Defects4C_bgcommit* across 1.1K projects for fine-tuning.

Due to resource constraints, we selected two popular base models, CodeLlama-7B-base and DeepSeek-coder-6.7B-base, for fine-tuning. We applied parameter-efficient fine-tuning using LoRA (Hu et al., 2021) with a rank of 8. The models were fine-tuned for 3 epochs with a learning rate of 2e-5. All experiments were conducted on 8 RTX A6000 GPUs, with a batch size of 8 per GPU.

## 5.3 PROMPT DESIGN

To interact with LLMs, we need to design appropriate prompts for experiments. Based on the three types of bugs/vulnerabilities, i.e., fixed in a single line, hunk, or function, as described in Section 4, we created corresponding prompts. These are illustrated in Figure 3, where parts 1, 2, and 3 correspond to function-level, hunk-level, and line-level bugs, respectively. In particular, for the prompt of single function bugs, we design the corresponding prompt to require the model to generate the complete function. Hence, the placeholder `Original Buggy Function` is a function, for example the placeholder `Error Message` in part 4 denotes the error information provided by the compiler based on the patch of last iteration. For the prompt of the single hunk and single line bugs, as the error statements are continuous, we mask them in the original function by the symbol >>>[INFILL]<<<

Table 3: Evaluating LLMs on *Defects4C* for single-round repair, where $k^* = 1$ marks pass@1 done with greedy-search decoding and pass@$k$ results with its corresponding temperature.

| Model | Size | $k^*{=}1$ | T=0.2 | | | T=0.8 | | |
|---|---|---|---|---|---|---|---|---|
| | | | $k=1$ | $k=10$ | $k=100$ | $k=1$ | $k=10$ | $k=100$ |
| GPT-4 | N/A | **9.0** | - | - | - | - | - | |
| GPT-35-Turbo | N/A | 8.5 | 7.9 | 13.5 | 19.5 | 7.1 | 20.0 | 38.9 |
| CodeLlama-Instruct | 7B | 2.5 | 3.3 | 11.1 | 24.9 | 4.8 | 20.5 | 45.7 |
| | 13B | 5.3 | 4.0 | 14.2 | 25.7 | 3.8 | 18.1 | 40.4 |
| | 34B | 4.0 | 3.6 | 12.1 | 25.7 | 3.2 | 14.7 | 35.9 |
| CodeLlama-Python | 7B | 0.0 | 0.1 | 1.2 | 4.5 | 0.8 | 6.2 | 22.5 |
| | 13B | 0.0 | 0.3 | 1.8 | 4.5 | 1.7 | 11.2 | 32.2 |
| | 34B | 0.0 | 0.3 | 2.2 | 6.9 | 1.2 | 8.8 | 29.8 |
| CodeLlama-Base | 7B | 0.0 | 0.0 | 0.0 | 0.0 | 0.2 | 2.1 | 14.3 |
| deepseek-coder-base | 6.7B | 0.4 | 0.3 | 1.0 | 3.7 | 0.9 | 6.8 | 25.7 |
| | 33B | 0.0 | 0.0 | 0.0 | 0.0 | 0.7 | 5.7 | 26.1 |
| deepseek-coder-instruct | 6.7B | 1.2 | 2.4 | 10.7 | 25.7 | 2.2 | 13.4 | 33.9 |
| Gemma | 7B | 0.0 | 0.4 | 3.0 | 11.0 | 0.8 | 6.6 | 26.9 |
| | 7B-Instruct | 0.0 | 0.8 | 5.1 | 14.7 | 0.9 | 6.1 | 22.9 |
| | Code7B | 0.0 | 0.0 | 0.0 | 0.0 | 0.0 | 0.2 | 1.2 |
| Magicoder-S-DS | 6.7B | 0.0 | 0.0 | 0.0 | 0.0 | 0.0 | 0.0 | 0.0 |
| Mixtral-8x7B-Instruct | 7B | 0.0 | 0.0 | 0.0 | 0.0 | 0.0 | 0.0 | 0.0 |
| phi-2 | 2.7B | 0.0 | 0.0 | 0.0 | 0.0 | 0.4 | 3.7 | 19.9 |
| Phind-CodeLlama | 34B | 6.1 | 5.4 | 18.6 | 34.7 | 4.8 | 20.6 | 38.4 |
| WizardCoder-Python | 7B | 0.0 | 0.2 | 1.1 | 3.7 | 0.4 | 3.4 | 18.8 |
| | 13B | 0.0 | 0.7 | 4.2 | 11.8 | 1.4 | 11.0 | 35.5 |
| | 34B | 4.4 | 5.2 | 13.0 | 21.2 | 5.5 | 23.0 | 45.1 |
| WizardCoder | 15B | 1.0 | 1.1 | 4.9 | 11.3 | 1.7 | 10.4 | 28.9 |
| | 33B | 0.0 | 0.0 | 0.0 | 0.0 | 0.2 | 1.9 | 10.3 |

Table 4: Evaluating LLMs on *Defects4C* for conversation-based repair where Pass denotes the number of bugs or vulnerabilities that the model can successfully repair, Avg.tries denotes the average tries of the successful repair. Due to the limited budget, the maximum number of repair attempts is set to 2 for GPT-4, and the remaining models are set to 10 by default.

| Model | Decoding | Defects4C_bug | | | | | | | | | | Defects4C_vul | | | | | | | | |
|---|---|---|---|---|---|---|---|---|---|---|---|---|---|---|---|---|---|---|---|---|
| | | Signature | | Sanitizer | | Memory Error | | Logic | | Pass/Sum | Signature | | Sanitizer | | Memory Error | | Logic | | Pass/Sum |
| | | Pass/Total | Avg.tries | Pass/Total | Avg.tries | Pass/Total | Avg.tries | Pass/Total | Avg.tries | | Pass/Total | Avg.tries | Pass/Total | Avg.tries | Pass/Total | Avg.tries | Pass/Total | Avg.tries | |
| GPT-4 | T=1.0 | 0/75 | 0.0 | 4/66 | 2.0 | 1/20 | 1.0 | 0/87 | 0.0 | 5/248 | **1**/11 | 2.0 | 0/6 | 0.0 | 0/72 | 0.0 | 0/13 | 0.0 | 1/102 |
| | greedy | 3/75 | 2.0 | 1/66 | 1.0 | 1/20 | 2.0 | 0/87 | 0.0 | 5/248 | **1**/11 | 2.0 | 0/6 | 0.0 | 3/72 | 1.3 | 0/13 | 0.0 | 4/102 |
| GPT-35-Turbo | T=1.0 | 8/75 | 1.7 | 4/66 | 3.0 | 3/20 | 3.7 | 4/87 | 2.7 | **27**/248 | 0/11 | 0.0 | 1/6 | 10.0 | 0/72 | 0.0 | 0/13 | 0.0 | 1/102 |
| | greedy | 7/75 | 2.0 | 4/66 | 3.0 | **5**/20 | 2.8 | 2/87 | 1.0 | 18/248 | 0/11 | 0.0 | **2**/6 | 4.5 | 2/72 | 8.5 | 0/13 | 0.0 | 4/102 |
| CodeLlama-Instruct-7B | T=1.0 | **9**/75 | 2.8 | 11/66 | 2.9 | 3/20 | 3.0 | 4/87 | 6.3 | **27**/248 | 0/11 | 0.0 | 0/6 | 0.0 | 0/72 | 0.0 | 0/13 | 0.0 | 0/102 |
| | greedy | 3/75 | 6.0 | 8/66 | 4.6 | 4/20 | 4.7 | 1/87 | 1.0 | 16/248 | 0/11 | 0.0 | 0/6 | 0.0 | 0/72 | 0.0 | **1**/13 | 9.0 | 1/102 |
| WizardCoder-Python-34B | T=1.0 | 0/75 | 0.0 | 0/66 | 0.0 | 0/20 | 0.0 | 1/87 | 1.0 | 1/248 | 0/11 | 0.0 | 0/6 | 0.0 | 0/72 | 0.0 | 0/13 | 0.0 | 0/102 |
| | greedy | 0/75 | 0.0 | 0/66 | 0.0 | 0/20 | 0.0 | 0/87 | 0.0 | 0/248 | **1**/11 | 8.0 | 0/6 | 0.0 | 0/72 | 0.0 | 0/13 | 0.0 | 1/102 |
| Gemma-Instruct-7B | T=1.0 | 0/75 | 0.0 | 1/66 | 1.0 | 0/20 | 0.0 | 0/87 | 0.0 | 1/248 | 0/11 | 0.0 | 0/6 | 0.0 | 1/72 | 3.0 | 0/13 | 0.0 | 1/102 |
| | greedy | 1/75 | 8.0 | 0/66 | 0.0 | 0/20 | 0.0 | 0/87 | 0.0 | 1/248 | 0/11 | 0.0 | 0/6 | 0.0 | 0/72 | 0.0 | 0/13 | 0.0 | 0/102 |
| Phind-CodeLlama-34B | T=1.0 | **9**/75 | 4.9 | 4/66 | 6.7 | 1/20 | 8.0 | 4/87 | 4.7 | 18/248 | 0/11 | 0.0 | **2**/6 | 1.0 | **5**/72 | 4.8 | 0/13 | 0.0 | **7**/102 |
| | greedy | 0/75 | 0.0 | 2/66 | 1.0 | 4/20 | 1.0 | 1/87 | 8.0 | 7/248 | 0/11 | 0.0 | 1/6 | 1.0 | 1/72 | 1.0 | 0/13 | 0.0 | 2/102 |
| deepseek-coder-33b-base | T=1.0 | 4/75 | 1.5 | 0/66 | 0.0 | 2/20 | 1.0 | 0/87 | 0.0 | 6/248 | 0/11 | 0.0 | 0/6 | 0.0 | 0/72 | 0.0 | 0/13 | 0.0 | 0/102 |
| | greedy | 0/75 | 0.0 | 0/66 | 0.0 | 0/20 | 0.0 | **6**/87 | 8.2 | 6/248 | 0/11 | 0.0 | 0/6 | 0.0 | 0/72 | 0.0 | 0/13 | 0.0 | 0/102 |

and provide these error statements by the placeholder `Masked Code Snippet` for the model to generate masked statements. An example is shown in part 5.

For single-round repair, we directly feed the prompts to the model. For conversation-based repair, the designed prompts serve as the initial input to the LLMs. After the model generates an output, the compiler evaluates it. If the output fails to pass the verification, the newly produced compilation error is appended to the prompt template to construct a new prompt for the next round of repair. For fine-tuning, we use the prompt without the compilation error, which is the same prompt as the single-round repair for the train.

# 6 EXPERIMENTAL RESULTS

## 6.1 SINGLE-ROUND REPAIR

The experimental results of different LLMs on *Defects4C* for single-round repair are presented in Table 3. Generally, we can observe that setting the temperature to 0.8 usually performs better than the temperature to 0.2, which indicates the improvement of the diversity in model output usually

Table 5: The repair performance compared with Defects4J. #Avg.tries represents the average number of attempts required, calculated as the ratio of successful repairs (Pass) to the total attempts (Total).

| Model | | Func | | Hunk | | Line | | #Avg.tries |
|---|---|---|---|---|---|---|---|---|
| | | #Pass/Total | Rate | #Pass/Total | Rate | #Pass/Total | Rate | |
| Defects4J (Xia & Zhang, 2024) | | - | 29.80 | - | 51.30 | - | 71.30 | - |
| GPT4 | T=1 | 1/46 | 2.17 | 2/179 | 1.12 | 7/125 | 5.60 | 2.86 |
| | greedy | 0/46 | 0.00 | 7/179 | 3.91 | 2/125 | 1.60 | 2.57 |
| GPT-3.5-Turbo | T=1 | 0/46 | 0.00 | 11/179 | 6.15 | 17/125 | 13.60 | 8.00 |
| | greedy | 0/46 | 0.00 | 9/179 | 5.03 | 13/125 | 10.40 | 6.29 |
| CodeLlama-Instruct-7B | T=1 | 0/46 | 0.00 | 10/179 | 5.59 | 17/125 | 13.60 | 7.71 |
| | greedy | 0/46 | 0.00 | 10/179 | 5.59 | 7/125 | 5.60 | 4.86 |
| WizardCoder-Python-34B | T=1 | 0/46 | 0.00 | 1/179 | 0.56 | 0/125 | 0.00 | 0.29 |
| | greedy | 0/46 | 0.00 | 1/179 | 0.56 | 0/125 | 0.00 | 0.29 |
| Gemma-Instruct-7B | T=1 | 0/46 | 0.00 | 1/179 | 0.56 | 1/125 | 0.80 | 0.57 |
| | greedy | 0/46 | 0.00 | 1/179 | 0.56 | 0/125 | 0.00 | 0.29 |
| Phind-CodeLlama-34B | T=1 | 2/46 | 4.35 | 12/179 | 6.70 | 11/125 | 8.80 | 7.14 |
| | greedy | 0/46 | 0.00 | 3/179 | 1.68 | 6/125 | 4.80 | 2.57 |
| deepseek-coder-33b-base | T=1 | 0/46 | 0.00 | 4/179 | 2.23 | 2/125 | 1.60 | 1.71 |
| | greedy | 0/46 | 0.00 | 6/179 | 3.35 | 0/125 | 0.00 | 1.71 |

contributes to better program repair. We can also find that as the number of $k$ increases, the success rate of repairs also improves. It is reasonable because increasing the number of generated outputs enhances the probability of correctly generating repair code.

Further analysis of different variants of the same model reveals that increasing model size does not necessarily lead to better repair accuracy. For instance, when the size of CodeLlama-Python increases from 7B to 13B, pass@100 improves from 22.4 to 32.2. However, with CodeLlama-Python 34B, pass@100 drops to 29.8. Similar trends are observed in WizardCoder-15B/33B and CodeLlama-Instruct. In contrast, some models, like deepseek-coder and WizardCoder-Python, show the opposite trend. This suggests that increasing model size does not guarantee improved performance; it is still dependent on the specific model and task. Moreover, several open-source models, such as Magicoder, perform poorly on *Defects4C*, despite excelling on popular datasets like HumanEval (Chen & Zaremba, 2021). Interestingly, the performance gap between open-source and closed-source models on *Defects4C* is less pronounced compared to their performance on other datasets (Chen & Zaremba, 2021; Liu et al., 2023). This indicates that *Defects4C*, collected from real-world projects, presents a more challenging testbed, further underscoring the value of the dataset.

### 6.2 CONVERSATION-BASED REPAIR

We then selected the best-performing models from Table 3 to conduct experiments on conversation-based repair, with the results presented in Table 4. Overall, we found that LLMs perform better at repairing *Defects4C_bug* than *Defects4C_vul*. For instance, LLMs were able to successfully repair 27 bugs, compared to only 7 vulnerabilities. We speculate that this difference may be due to the increased complexity of vulnerabilities, which makes them more challenging for LLMs to address. However, the results show that LLMs were able to repair only 27 out of 248 bugs and 7 out of 102 vulnerabilities. This low performance highlights the significant room for improvement in LLMs' ability to repair C/C++ defects.

Additionally, we observed that for *Defects4C_bug*, GPT-4 successfully repaired 5 bugs, which is lower than GPT-3.5's performance (i.e., 27 bugs). However, for *Defects4C_vul*, GPT-4 handled 5 vulnerabilities, outperforming GPT-3.5, which repaired only 4. It's important to note that we limited the maximum number of repair attempts for GPT-4 to 2 due to budget constraints, while other models had up to 10 attempts. We believe that GPT-4 could achieve higher repair accuracy with more repair attempts. Furthermore, we found that setting the model's temperature to 1.0 generally resulted in better repair accuracy compared to using greedy search decoding. Lastly, apart from GPT-4 and GPT-3.5, open-source models performed poorly even in conversation-based repair. For example, WizardCoder and Gemma were able to repair only 1 bug or vulnerability in both *Defects4C_bug* and *Defects4C_vul*. This suggests that while these open-source models may excel in certain tasks or datasets, their generalizability remains limited.

Table 6: Comparative Results of LLMs With and Without Fine-Tuning.

| Model | Finetune | Greedy | T=0.2 | | | T=0.8 | | |
|---|---|---|---|---|---|---|---|---|
| | | | $k=1$ | $k=10$ | $k=100$ | $k=1$ | $k=10$ | $k=100$ |
| CodeLlama-7B-Base | ✗ | 0.00 | 0.00 | 0.00 | 0.00 | 0.22 | 2.10 | 14.29 |
| | ✓ | 0.41 | 0.25 | 0.92 | 2.86 | 0.44 | 3.72 | 20.41 |
| CodeLlama-7B-Instruct | ✗ | 2.45 | 3.31 | 11.07 | 24.90 | 4.81 | 20.51 | 45.71 |
| | ✓ | 4.08 | 4.26 | 9.30 | 17.14 | 4.92 | 20.99 | 46.94 |
| Deepseek-Coder-6.7B-Base | ✗ | 0.41 | 0.33 | 0.96 | 3.67 | 0.87 | 6.83 | 25.71 |
| | ✓ | 0.41 | 0.19 | 0.45 | 0.82 | 0.24 | 1.58 | 5.31 |
| Deepseek-Coder-6.7B-Instruct | ✗ | 1.22 | 2.42 | 10.65 | 25.71 | 2.16 | 13.36 | 33.88 |
| | ✓ | 3.27 | 3.74 | 10.49 | 20.82 | 3.87 | 18.41 | 41.22 |

**Performance Comparison with APR on Defects4J.** We further compare the performance with existing state-of-the-art APR on Defects4J. Specifically, we select the conversation-based ChatRepair (Xia & Zhang, 2024) and directly report the repair success rate on Defects4J in the categories of *Line*, *Hunk* and *Function* from their original paper. ChatRepair is the first work that adopts GPT-3.5 in a conversational manner for bug fixing. The comparison results are presented in Table 5, where the first row is the state-of-the-art repair performance from ChatRapir (Xia & Zhang, 2024) on Defects4J. Compared with the repair success rate on Defects4J, the performance in repairing C/C++ bugs and vulnerabilities is significantly lower, underscoring the inherent challenges in fixing C/C++ faults and the pressing need for more specific repair methods. We also present a case study in Appendix C, showcasing examples of both successful and failed repairs by LLMs.

### 6.3 FINETUNING-BASED REPAIR

The fine-tuned results are presented in Table 6. The second column, *Finetune*, indicates whether the model has been fine-tuned with *Defects4C_bgcommit*, where ✗ represents results from the pre-trained model (listed here for comparison purposes), and ✓ represents results with LoRA-based fine-tuning. Overall, we observe that fine-tuning does not always lead to improved performance and, in some cases, can even reduce performance.

For various versions of CodeLlama, fine-tuning generally enhances repair capabilities. However, for Deepseek, performance inconsistency increases. Specifically, fine-tuning decreases repair performance in the base version of Deepseek, whereas in the instruct version, fine-tuning improves repair accuracy, particularly when the temperature is set to 0.8. Additionally, the results show that setting the temperature to 0.8 typically yields better repair performance compared to a temperature of 0.2 or using greedy decoding during fine-tuning.

These experimental findings suggest that while fine-tuning shows some promise, it may not always be effective when applied directly. This highlights the need for more advanced fine-tuning methods to further improve C/C++ program repair.

## 7 CONCLUSION AND FUTURE WORK

In this paper, we introduced *Defects4C*, a comprehensive and high-quality C/C++ defect benchmark that significantly advances the evaluation and fine-tuning of LLM-based automated program repair techniques. Our dataset addresses a major gap in the field by providing a large-scale resource specifically designed for C/C++ faults. Through extensive experiments on pre-trained models and fine-tuned models, we uncovered several key findings. Specifcially, our evaluation of pre-trained LLMs revealed a notable performance gap when handling C/C++ faults compared to their effectiveness in Java-based benchmarks such as Defects4J. The preliminary results further show that direct fine-tuning is not always effective. While the results show some promise, they still fall short of acceptable levels.

Our work opens several avenues for future research based on our dataset. One promising direction is to improve the prompts provided to LLMs for repair tasks. Researchers could leverage static analysis tools or other C/C++-specific techniques to provide more detailed feedback (e.g., memory safety, undefined behavior, or specific compilation errors) in the prompt, enabling the model to generate higher-quality repairs. Another line of future work lies in improving the fine-tuning process itself. For example, we could select high-quality data from *Defects4C_bgcommit*, employ data augmentation or add more dynamic execution information for further boosting model performance.

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

Table 7: The error distribution across different projects.

| Project Name | Bugs/vulnerabilities | Line | Hunk | Function | Test Cases (avg.) |
|---|---|---|---|---|---|
| ARMmbed/mbedtls | 1 | 0 | 0 | 1 | 1.0 |
| awslabs/aws-c-common | 1 | 1 | 0 | 0 | 2.0 |
| bblanchon/ArduinoJson | 1 | 0 | 1 | 0 | 2.0 |
| CauldronDevelopmentLLC/cbang | 1 | 0 | 1 | 0 | 20.0 |
| curl/curl | 1 | 0 | 1 | 0 | 2.0 |
| DaveGamble/cJSON | 1 | 0 | 0 | 1 | 2.0 |
| dlundquist/sniproxy | 1 | 1 | 0 | 0 | 3.0 |
| DynamoRIO/dynamorio | 1 | 0 | 0 | 1 | 2.0 |
| mdadams/jasper | 1 | 0 | 1 | 0 | 1.0 |
| mongodb/mongo-c-driver | 1 | 0 | 1 | 0 | 1.0 |
| OpenIDC/cjose | 1 | 1 | 0 | 0 | 7.0 |
| PCRE2Project/pcre | 1 | 0 | 1 | 0 | 1.0 |
| SOCI/soci | 1 | 1 | 0 | 0 | 9.0 |
| redis/hiredis | 1 | 1 | 0 | 0 | 1.0 |
| redis/redis | 1 | 1 | 0 | 0 | 2.0 |
| VirusTotal/yara | 1 | 0 | 0 | 1 | 1.0 |
| webmproject/libvpx | 1 | 1 | 0 | 0 | 2.0 |
| wez/atomicparsley | 1 | 0 | 1 | 0 | 1.0 |
| Yeraze/ytnef | 1 | 1 | 0 | 0 | 1.0 |
| yhirose/cpp-peglib | 1 | 0 | 1 | 0 | 2.0 |
| lua/lua | 2 | 1 | 1 | 0 | 2.5 |
| skypjack/entt | 2 | 1 | 1 | 0 | 4.0 |
| uncrustify/uncrustify | 2 | 1 | 1 | 0 | 2.0 |
| uriparser/uriparser | 2 | 2 | 0 | 0 | 3.0 |
| jqlang/jq | 2 | 0 | 2 | 0 | 1.0 |
| CLIUtils/CLI11 | 3 | 1 | 2 | 0 | 4.3 |
| facebook/rocksdb | 3 | 3 | 0 | 0 | 3.7 |
| libevent/libevent | 3 | 0 | 2 | 1 | 1.3 |
| nanomsg/nng | 3 | 0 | 3 | 0 | 1.0 |
| libgd/libgd | 4 | 1 | 2 | 1 | 2.8 |
| sqlite/sqlite | 4 | 2 | 1 | 1 | 2.0 |
| zeromq/libzmq | 4 | 1 | 1 | 2 | 5.0 |
| apache/arrow | 9 | 6 | 3 | 0 | 9.1 |
| nginx/njs | 10 | 4 | 5 | 1 | 2.5 |
| KhronosGroup/SPIRV-Tools | 12 | 7 | 4 | 1 | 2.3 |
| fmtlib/fmt | 14 | 8 | 6 | 0 | 2.1 |
| CESNET/libyang | 15 | 10 | 4 | 1 | 3.8 |
| php/php-src | 18 | 4 | 10 | 4 | 2.4 |
| danmar/cppcheck | 32 | 25 | 6 | 1 | 1.8 |
| the-tcpdump-group/tcpdump | 43 | 20 | 8 | 15 | 2.9 |
| llvm/llvm-project | 143 | 20 | 109 | 14 | 2.7 |
| Total | 350 | 125 | 179 | 46 | 3.1 |

# A MORE DETAILS OF *Defects4C*

We provide more details of our *Defects4C* as follows.

## A.1 ERROR DISTRIBUTION

The number of erroneous functions each project has is presented in Table 7. We also present the average test cases on *Defects4C*. The average test cases of *Defects4C*: 3.1, are higher than Defects4J: 2.4.

## A.2 DETAILS OF ERROR CATEGORIES

Note that the categories are mainly inspired from OctoPack (Muennighoff et al., 2023) and Magicoder (Wei et al., 2023). The 7 categories are designed specific to the collected bugs, which cover the vast majority of different applications and most of them are consistent with Magicoder. A detailed introduction of each error category from Table 2 is presented as follows:

***Signature.*** 75 bugs and 11 vulnerabilities are categorised as `Signature`, whose modifications only involve code elements within a single line of code i.e., LoC, for instance, wrong function name or variable. These errors are often relatively easy to fix yet usually require a certain level of contextual understanding to modify and correctly use the appropriate calling function or variable. This category is further divided into four subcategories based on their root causes, which are `Incorrect Function Usage`, `Fault Input Type`, `Incorrect Function Return Value` and `Incorrect Variable Usage`, respectively. We provide a detailed introduction to these subcategories.

- `Incorrect Function Usage`. This bug category frequently entails the misuse of functions, encompassing both third-party library functions and internal methods within code objects. Remedying these bugs typically involves substituting the fault function call with the correct one. Such corrections demand a comprehensive understanding of the overall software project, as well as a deep semantic grasp of the logic underlying the employed methods.

- `Fault Input Type`. In statically typed languages, the accurate specification of variables and return value is crucial. Bugs in this category frequently arise from incorrect variable type assignments within the code, resulting in unforeseen errors.

- `Incorrect Function Return Value`. During our analysis, it was observed that a significant number of bugs stem from improper settings of return values in specific condition structures or function calls. Rectifying these bugs typically necessitates altering the return value to align with the correct code logic. This correction process demands not only an understanding of the code's context but also a comprehensive semantic comprehension of the pertinent functions or conditional logic.

- `Incorrect Variable Usage`. These bugs bear a resemblance to the `Incorrect Function Usage` bugs; however, they primarily involve the improper use of variables instead of functions. The erroneously used variable might appear independently in a code statement or within a function call. Consequently, these bugs, compared to bugs in the first subcategory, are often more complex and challenging to rectify due to their increased flexibility in occurrence.

For bugs categorized in **Signature**, while generally simpler to rectify, necessitate a substantial level of contextual understanding for accurate modification, particularly in selecting and utilizing the appropriate calling functions or variables.

***Sanitizer.*** This category refers to bugs or vulnerabilities whose fix locations only involve the conditional logic within a LoC, such as changes in the value domain within an *if* condition. The modifications for fixing these bugs are generally minimal (e.g., changing the conditional check from $<$ to $\leq$). However, these bugs can often lead the software into incorrect operational logic under specific input conditions. Our analysis identified 66 `Sanitizer` bugs and 6 vulnerabilities, which can be categorized under the root cause of `Control Expression Error`. The root cause of these bugs can be classified as *Control Expression Error*. The modifications required to fix these types of bugs are usually minimal. However, such bugs can lead to incorrect operational logic in the software under certain input conditions. For instance, in the *cppcheck* project's *CheckCondition::alwaysTrueFalse* method (Danmar/cppcheck, 2007), a bug was identified where the if condition erroneously employed the logical *AND* operator instead of the logical *OR* operator. This error resulted in the generation of false positive results.

***Memory Error.*** We categorize the bugs/vulnerabilities that would trigger the fault behaviours of memory as a separate category, as in memory-unsafe languages like C and C++, there are many bugs related to memory that can lead to serious consequences (e.g., CVE-2018-8301 (Corporation, 2018) corrupt memory usage leading to remote command execution). In *Defects4C*, we identified 20 memory-related bugs and 72 vulnerabilities and summarized them into three subcategories, namely `Null Pointer Dereference`, `Uncontrolled Resource Consumption`, and `Memory Overflow`.

- `Null Pointer Dereference`. These vulnerabilities could refer to CWE-476 (NULL Pointer Dereference, 2005), which occurs in the software when a pointer is used without properly checking if its value is NULL, leading to program crashes or other undefined behaviours.

- `Uncontrolled Resource Consumption`. These vulnerabilities correspond to CWE-400 (Uncontrolled Resource Consumption, 2005), which can lead to resource exhaustion, thereby impacting the system's performance or stability. Notably, 45.0% of the memory-related bugs fall into this category.

- `Memory Overflow`. These types of bugs mainly relate to memory overflow vulnerabilities (e.g., CWE-122 (Heap-based Buffer Overflow, 2005), CWE-121 (Stack-based Buffer Overflow, 2005), etc.). Such bugs often involve the leakage of sensitive memory information and pose serious security risks.

***Logic Organization.*** Among 87/13 bugs/vulnerabilities involving multiple LoC modifications, we found that these bugs are often related to the handling and organization of code logic. They can

be categorized into two subcategories: `Improper Condition Organization` and `Wrong Function Call Sequence`.

- `Improper Condition Organization`. There are 67/11 bugs/vulnerabilities classified into this subcategory, which can correspond to CWE-391 (Unchecked Error Condition, 2005). These bugs often involve improper wrappings of condition logic. For instance, in the *xml_print_opaq_open* method of the *libyang* project (CESNET/LibYang, 2017b), the lack of namespace checking when calling the *xml_print_ns_opaq* function would lead to the fault printed namespace result. The corresponding bug fix logic often involves adding/removing a nested structure of conditional code (e.g., an if-else pair) within the existing code block to guide the code towards the correct logic.

- `Wrong Function Call Sequence`. The root cause of this bug category could align with CWE-691 (Insufficient Control Flow Management, 2007). Such bugs typically arise from incorrect code-calling logic. Consequently, the bug fixes of these bugs involve relocating one or more complete code blocks to different locations, without altering the content within these blocks (LLVM/LLVM-project, 2020).

## A.3 EXPERIMENTS CONFIGURATION

**Docker and Compiler Configuration.** All project within our system is furnished with the same Docker file, thereby establishing a uniform execution environment, and there is no need to establish an individual for each bug which is time exhausted. All bugs can be reproduced within this Docker container, as our project's initial purpose is to make reproducing and compiling as quick as possible to obtain the final test results, especially for LLM-based massive compilation tasks like passrate@100. Both Docker configurations are build for Ubuntu 20.04-x86_64, accommodating either clang-16 or GCC-9 as the designated compilers. Specifically, projects such as *awslabs/aws-c-common*, *DynamoRIO/dynamorio*, *llvm/llvm-project*, *skypjack/entt*, *KhronosGroup/SPIRV-Tools*, and *facebook/rocksdb* are compiled with GCC-9.

**Compilation Flags and Dependency Management.** Compilation flags are derived from the CI script or CMakefile.txt from each project's GitHub. In terms of compilation variables, like *-DARROW_BUILD_SHARED=on*, uniformity is rigorously kept between the buggy-commit and patch-commit stages of a bug, ensuring replicated and stability. Dependencies are split into system-level and user-defined. System-level libraries are installed during the Docker image building phase or will be integrated into the Docker image upon publishing. User-defined libraries are installed once during the project's initial phase and do not need to be reinstalled during subsequent compilations. It is noteworthy that each identified bug can have specific library requirements if require a specified dependency or compilation flag, with more details provided in each project's meta file, such as the *project.json* file.

**Unit Test Reporting.** The build tool used across the *Defects4C_bug* and *Defects4C_vul* projects is *CMake* version 2.6, with *Ninja* employed for building, and *ctest* used to generate JUnit-style Unit Test reports. Test cases are extracted from these reports by navigating to any leaf node labeled "testcase." Test error messages are taken from the test report, while most compilation errors are gathered from the CMake error report. Note that some projects use the native Unix build tool, such as *configure* or *autogen*; please refer to each project's repository for details on how to build, execute, and report. For example, the project *llvm/llvm-project* equipped with its own test frameworks, we follow its respective test pipelines, such as *llvm-lit*. For the lots of projects, the testing process is executed through the *ctest* CLI interface.

**Computer Resource.** Specifically, for the cpu task, like compilation, we conducted our experiments using a machine equipped with an 80-core Intel Xeon E5 Processor, 256GB of memory. All experiments related to GPT were conducted using the OpenAI official API, i.e., GPT-3.5-turbo-0125 and the gpt-4-turbo-preview at March 2024. For experiments involving open-source models such as WizardCoder, CodeLlama etc., the opensource framework vLLM (Kwon et al., 2023) was utilized. These models were deployed on eight NVIDIA RTX A6000 GPUs.

## B MORE DETAILS OF CONVERSATION-BASED REPAIR SETUP

For the conversation-based repair, we delineate the details of our experimental settings. Here, we introduce two hyperparameters, $m$ and $n$, representing the maximum number of repair attempts and the maximum conversation length in each attempt, with values of m and n being set as 10 and 3, respectively. Specifically, one repair attempt consists of three continuous conversations. The aim is not only to resolve failures but also to evolve its performance automatically by iteratively investigating failure points during the same attempt phase.

The following illustrates a conversation-based prompt, given a buggy function $F_n$ and its error message $M_n$ represent the $n^{th}$ conversation in one attempt. At the beginning of this attempt, the $F_0$ and $M_0$ are extracted from *Defects4C* concatenated to construct the prompt. At the patch's verification phase, for example, the first conversation phase, the LLM outputs a patch and evaluates with corresponding Unit Test cases getting an Error Message $M_1$; if it can pass all the test cases, this patch is considered plausible, then the conversation stops, and the repair process ends, otherwise, this patch is invalid, and we will updated prompt format with its error message $M_1$ and updated buggy function $F_1$, to build a new prompt for the continuous conversation. Following this rationale, after three iterations of conversation, the repair will reset to the initial prompt and start a new attempt loop.

## C CASE STUDY

In this section, we will choose four representative instances of bugs to serve as case studies for conversation-based repair tasks implemented on the two models, including GPT-3.5-Turbo (GPT-3.5) and Phind-CodeLlama-34B (Phind34B). These instances encompass scenarios wherein both models are effecting repairs, only one model demonstrates efficacy, and neither model exhibits repair capability.

**Successful repair by both GPT-3.5 and Phind34B.** To describe this kind of bug, we take the bug (CESNET/LibYang, 2017a) in the function `lyjson-number` as an example, as shown in Table 8. This bug falls into the **Sanitizer** category. To fix it, the expression to the right of the < operand must be changed from `exponent` to `(exponent - minus)`. Both GPT-3.5 and Phind34B understand the buggy semantics and successfully output plausible patches to correct the bug. For Phind34B, the patch it generates is identical to the one provided by the developers. However, the patch generated by GPT-3.5, `(exponent - 1)`, is also semantically equivalent because, at the beginning of the function, the variable `minus` is initialized to `1` and is never modified afterward. As a result, both patches are semantically equivalent and allow the function to pass all the test cases.

---

**Listing 1** File src/json.c

```
 uint8_t minus = 0;
 if (in[offset] == '-') { minus = 1; }
 ...
 num_len = exponent + e_val;
- } else if ((size_t)labs(e_val) < exponent) {
+ } else if ((size_t)labs(e_val) < (exponent - minus)) {
 num_len = exponent + 1;
 dp_position = exponent + e_val;
```

---

| Model | Patch | Status |
|---|---|---|
| Bug | - } else if ((size_t)labs(e_val) < exponent) { | Fail |
| Developer | +} else if ((size_t)labs(e_val) < (exponent - minus)) { | Pass |
| Phind34B($a_1$) | +} else if ((size_t)labs(e_val) < exponent) { | Fail |
| Phind34B($a_7$) | +} else if ((size_t)labs(e_val) < (exponent **- minus**)) { | Pass |
| GPT-3.5($a_1$) | +} else if ((size_t)labs(e_val) < exponent) { | Fail |
| GPT-3.5($a_3$) | +} else if ((size_t)labs(e_val) < exponent **- 1**) { | Pass |

Table 8: The showcase of the bug's patch from different models with the first and last attempt is summarized. Here, $a_m$ represents the number of the $m^{th}$ attempt. The Status column denotes the outcome of patch verification as assessed through the associated Unit Test. The developer row stands for the post-commit, emphasizing real-world patches that successfully pass this Unit Test. Listing 1 illustrates the differences between the bug and the developer's patch across two commits.

**Successful repair by only GPT-3.5.** We select a bug (Common, 2016a) from project *aws-c-common*, as illustrated by Table 9, which can only be repaired by GPT-3.5 but not Phind34B. This bug belongs to **Signature: Fault Input Type**, in order to correct it, the type of the first parameter in function `s_base64_get_decoded_value` should be modified from `char` to `unsigned char`. In this case, GPT-3.5 can generate as same patch as the developers provide but Phind34B fails to output a plausible patch. Moreover, Phind34B is not able to comprehend the root cause that triggers the bug even if we have provided the information of fault localization. With this hint, Phind34B still insists the bug is triggered by the other elements and modifies code snippets somewhere else.

**Listing 2** File source/encoding.c

```
- static inline int s_base64_get_decoded_value(char to_decode, ...) {
+ static inline int s_base64_get_decoded_value(unsigned char to_decode,
↪   ...) {
    uint8_t decode_value = BASE64_DECODING_TABLE[(size_t)to_decode];
    if (decode_value != 0xDD && (decode_value != BASE64_SENTIANAL_VALUE
↪   || allow_sentinal)) {
        *value = decode_value;
        return AWS_OP_SUCCESS;
    }
    return AWS_OP_ERR;
}
```

| Model | Patch | Status |
|---|---|---|
| Bug | +static inline int s_base64_get_decoded_value( char to_decode,...) { | Fail |
| Developer | +static inline int s_base64_get_decoded_value(**unsigned** char to_decode, ...) { | Pass |
| Phind34B($a_1$) | N/A | Fail |
| Phind34B($a_{10}$) | +static inline int s_base64_get_decoded_value (char to_decode, ...) { | Fail |
| GPT-3.5($a_1$) | +static inline int s_base64_get_decoded_value(char to_decode,...) { | Fail |
| GPT-3.5($a_2$) | +static inline int s_base64_get_decoded_value(**unsigned** char to_decode, ...) { | Pass |

Table 9: The showcase for only one model demonstrates efficacy, N/A represents no patch can be retrieved from LLM output, and the function name and signatures are omitted for space limit. The Listing 2 shows two commits' differences.

**Successful repair by only Phind34B.** We select a bug (Common, 2016b) from the project *apache/arrow*, as illustrated in Table 10, which can be repaired by Phind34B only. This bug falls under the **Signature: Incorrect Function Usage** category. To correct it, the type of the first parameter in the function `min\_args` should be modified from `1` to `0`. In this case, Phind34B generates the same patch as the developers at its second round, but GPT-3.5 fails to output a plausible patch. Additionally, GPT-3.5 incorrectly identifies other elements as the cause and modifies code snippets elsewhere, even after 10 rounds of prompting

**Listing 3** File cpp/src/arrow/compute/function.h

```
 static Arity Ternary() { return Arity(3, false); }
   /// \brief A function taking a variable number of arguments
- static Arity VarArgs(int min_args = 1) { return Arity(min_args, true);
↪   }
+   /// \param[in] min_args the minimum number of arguments required when
+   /// invoking the function
+ static Arity VarArgs(int min_args = 0) { return Arity(min_args, true);
↪   }
 explicit Arity(int num_args, bool is_varargs = false)
       : num_args(num_args), is_varargs(is_varargs) {}
```

**Failed repair by GPT-3.5 and Phind34B.** Given the low successful repair rate of LLMs on the *Defects4C*, this kind of bug constitutes a substantial proportion of the dataset. In this section, we select bugs that are unable to be repaired in either GPT-3.5 or Phind34B, we take

| Model | Patch | Status |
|---|---|---|
| Bug | -static Arity VarArgs(int min_args = 1)... { | Fail |
| Developer | +static Arity VarArgs(int min_args = **0**)... { | Pass |
| Phind34B($a_1$) | +static Arity VarArgs(int min_args)... { | Fail |
| Phind34B($a_2$) | +static Arity VarArgs(int min_args = **0**)... { | Pass |
| GPT-3.5($a_1$) | -static Arity VarArgs(int min_args = 1)... { | Fail |
| GPT-3.5($a_{10}$) | -static Arity VarArgs(int min_args = 1)... { | Fail |

Table 10: The showcase for only one model demonstrates efficacy and the function name and signatures are omitted for space limit. The Listing 3 shows two commits' differences.

this bug (Nanomsg/nng, 2018), selected from project *nng*, as an example in Table 11. This bug exists in function `nni_chunk_insert`, belonging to category **Memory Error: Uncontrolled Resource Consumption**, in which the identifier `ch->ch_ptr` should be substituted by `ch->ch_buf`. Actually, both Phind34B and GPT-3.5 have made many attempts to repair the bug, but none of the patches work. Below are several patches that have been generated with a high frequency of occurrence: 1. The third parameter in callee function `memmove` is replaced by `ch->ch_len - len`. 2.The third parameter in callee function `memmove` is replaced by `ch->ch_len - (ch->ch_ptr - ch->ch_buf))`. However, both of them are far away from the correct patch provided by developers. But we find an interesting patch that only appears once among all the patches generated by GPT-3.5 under $T$ set as 1, this patch tells adding an additional code line `ch->ch_ptr = ch->ch_buf;` behind the callee function `memmove`. As we can see, the keyword `ch->ch_buf` has appeared, and is also assigned to `ch->ch_ptr`, it's a partially correct patch. We think if the number of maximum repair attempts increases, this bug might be repaired, and more bugs that Phind34B/GPT-3.5 can generate partially correct patches will also be successfully repaired.

**Listing 4** File src/core/message.c

```
 static int nni_chunk_insert(nni_chunk *ch, ...)
 {
    ...
    ch->ch_ptr -= len;
 } else if ((ch->ch_len + len) <= ch->ch_cap) {
-    memmove(ch->ch_ptr + len, ch->ch_ptr, ch->ch_len);
+    memmove(ch->ch_buf + len, ch->ch_ptr, ch->ch_len);
 } else if ((rv = nni_chunk_grow(ch, 0, len)) == 0) {
    ch->ch_ptr -= len;
 }
    ...
 return (0);
 }
```

| Model | Patch | Status |
|---|---|---|
| Bug | -memmove(ch->ch_ptr + len, ch->ch_ptr, ch->ch_len); | Fail |
| Developer | +memmove(ch->**ch_buf** + len, ch->ch_ptr, ch->ch_len); | Pass |
| Phind34B($a_1$) | +memmove(ch->ch_ptr + len, ch->ch_ptr, ch->ch_len);
+ ch->ch_ptr -= len;
+ } else if ((rv = nni_chunk_grow(ch, 0, len)) == 0) {
+ ch->ch_ptr -= len;
+ } | Fail |
| Phind34B($a_{10}$) | N/A | Fail |
| GPT-3.5($a_1$) | +memmove(ch->ch_ptr + len, ch->ch_ptr, ch->ch_len); | Fail |
| GPT-3.5($a_{10}$) | +memmove(ch->ch_ptr + len, ch->ch_ptr, ch->ch_len); | Fail |

Table 11: The showcase for none of the models can be efficacy, N/A represents no patch that can retrieve from LLM output, and the Listing 4 shows the difference between the bug and the developer (real-world)'s patch across two commits.

## D  MODEL LIST

For our evaluation, we utilized the `GPT-3.5-turbo` and `GPT-4` models as of March 26, 2024. The HuggingFace URLs for the evaluated models are detailed in Table 12.

Table 12: Models and HuggingFace URLs

| Model Name | HuggingFace URL |
|---|---|
| CodeLlama Instruct (7B) | https://huggingface.co/codellama/CodeLlama-7b-Instruct-hf |
| CodeLlama Instruct (13B) | https://huggingface.co/codellama/CodeLlama-13b-Instruct-hf |
| CodeLlama Instruct (34B) | https://huggingface.co/codellama/CodeLlama-34b-Instruct-hf |
| CodeLlama Python (7B) | https://huggingface.co/codellama/CodeLlama-7b-Python-hf |
| CodeLlama Python (13B) | https://huggingface.co/codellama/CodeLlama-13b-Python-hf |
| CodeLlama Python (34B) | https://huggingface.co/codellama/CodeLlama-34b-Python-hf |
| CodeLlama Base (7B) | https://huggingface.co/codellama/CodeLlama-7b-hf |
| DeepSeek Base (6.7B) | https://huggingface.co/deepseek-ai/deepseek-coder-6.7b-base |
| DeepSeek Base (33B) | https://huggingface.co/deepseek-ai/deepseek-coder-33b-base |
| DeepSeek Instruct (6.7B) | https://huggingface.co/deepseek-ai/deepseek-coder-6.7b-instruct |
| DeepSeek Instruct (33B) | https://huggingface.co/deepseek-ai/deepseek-coder-33b-instruct |
| Gemma (7B) | https://huggingface.co/google/gemma-7b |
| Gemma (7B-Instruct) | https://huggingface.co/google/gemma-7b-it |
| Gemma (Code7B) | https://huggingface.co/TechxGenus/CodeGemma-7b |
| Magicoder-S-DS (6.7B) | https://huggingface.co/ise-uiuc/Magicoder-S-DS-6.7B |
| Mistral-8x7B-Instruct (8X7B) | https://huggingface.co/mistralai/Mixtral-8x7B-Instruct-v0.1/ |
| Phi-2 (2.7B) | https://huggingface.co/microsoft/phi-2 |
| Phind-CodeLlama (34B) | https://huggingface.co/Phind/Phind-CodeLlama-34B-v2 |
| WizardCoder-Python (7B) | https://huggingface.co/WizardLM/WizardCoder-Python-7B-V1.0 |
| WizardCoder-Python (13B) | https://huggingface.co/WizardLM/WizardCoder-Python-13B-V1.0 |
| WizardCoder-Python (34B) | https://huggingface.co/WizardLM/WizardCoder-Python-34B-V1.0 |
| WizardCoder (15B) | https://huggingface.co/WizardLM/WizardCoder-15B-V1.0 |
| WizardCoder (33B) | https://huggingface.co/WizardLM/WizardCoder-33B-V1.1 |

## E  SOURCE CODE

The *Defects4C* source code and instructions can be obtained from the website[3], which includes the source code for *Defects4C*, the source code for experiments, and the data generated by inference with VLLM, allowing for easy reproduction of results.

---

[3]https://sites.google.com/view/anonymous-defects4c

