# OpenReview forum: "Defects4C: Benchmarking C/C++ Faults to Assess LLM-Based Program Repair"
_ICLR.cc/2025/Conference — Submitted to ICLR 2025_

### Official Review · Reviewer_fJ2S · 2024-10-22

**Soundness:** 2
**Presentation:** 2
**Contribution:** 1
**Rating:** 3
**Confidence:** 5

**Summary:**

This paper proposes a defect dataset for C/C++ languages.

**Strengths:**

+ This could be a valuable dataset

**Weaknesses:**

- Overall, I think this paper is a dataset paper that could fit MSR dataset track well. However, it would not be qualified for a ICLR technical paper cause it contais little technical contribution.

- The description of the collection process of the vulnerability dataset is rather vague. I did not clearly understand how it was constructed. From the Introduction, it seems that this sub-dataset is directly reused from the official CVE database. Later, in the human annotation part, it seems that it was collected by humans, which makes me very confusing. Another point is that 248 bugs were chosen from 3,785 candidates, but 102 vulnerabilities were chosen from 102 condidates. What leads to such a big difference in this ratio?

- The manual annotation leads to Cohen’s Kappa values like 0.48 and 0.60 which are very low. This means that the annotation quality is rather low.

- In the annotation process, CWE types were used to help participants decide whether each commit is bug-related. It is well known that bugs and vulnerabilities are different concepts. Why use this type of information for bug identification?

- The prompts used for evaluation are simply wrong. Before fixing a bug, we would never know how many code entities we need to modify. That means what we can do is only providing a unified prompt for fixing. We cannot ask LLMs to perform modifications at a specific level because we do not have this prior knowledge. The experiments conducted in this study were thus totally wrong.

**Questions:**

1. What is the technical contribution of this paper?

2. Why we can use prompts of different levels for fixing bugs?

---

> ### Author Response · Authors · 2024-11-24
>
> We believe the reviewer has significantly misunderstood aspects of our paper, including the key content reading, the scope of ICLR, and foundational concepts in the program repair domain. While we appreciate the time taken to review our work, some comments suggest a lack of careful reading and understanding of the paper’s contributions. To be frank, we are disappointed by the reviews provided thus far but have made every effort to clarify these misunderstandings and address all concerns comprehensively.
>
> \
> We respectfully request that the reviewer reevaluate the paper with a fair and unbiased perspective. Thank you!
>
>
>
>
> >Q: Overall, I think this paper is a dataset paper that could fit MSR dataset track well. However, it would not be qualified for a ICLR technical paper cause it contains little technical contribution and What is the technical contribution of this paper?
>
> Please refer to the ICLR 2025 Call for Papers: https://iclr.cc/Conferences/2025/CallForPapers, which explicitly defines the conference's scope to include topics such as datasets and benchmarks.
>
> We believe the reviewer may have some misunderstanding or bias regarding the research topics addressed in our paper. As summarized in our general response, our contributions include a general bug collection framework, a high-quality dataset, and a large-scale empirical study. These contributions are highly relevant to the **dataset and benchmark** scope outlined by ICLR.
>
> We respectfully request the reviewer to reevaluate the paper from the perspective of the dataset and its alignment with the conference’s stated scope.
>
> >Q: The description of the collection process of the vulnerability dataset is rather vague. I did not clearly understand how it was constructed. From the Introduction, it seems that this sub-dataset is directly reused from the official CVE database. Later, in the human annotation part, it seems that it was collected by humans, which makes me very confusing. Another point is that 248 bugs were chosen from 3,785 candidates, but 102 vulnerabilities were chosen from 102 condidates. What leads to such a big difference in this ratio?
>
> We would like to clarify the dataset collection process and address your concerns.
>
> 1. Data Sources:
>     - We collect bugs (248) and vulnerabilities (102) from two primary sources:
>       - **GitHub real-world repositories** (see Introduction, lines 88–97).
>       - **The official CVE database** (see Introduction, lines 99–103).
>     - These sources have been described in separate paragraphs in the Introduction to clearly distinguish their origins.
>     - We name any samples from these two sources as defects4C_bug and defects4C_vul, respectively. (see Introduction, lines 081-087).
>
> 2. Collection and Annotation Process:
>    Both bugs and vulnerabilities follow a similar pipeline for verification and quality control, as illustrated in Figure 1 and detailed in Section 3:
>
>    - **Pipeline Steps**:
>      - Availability Validation
>      - Single-function filtering
>      - Test case filtering
>      - Unit test matching
>      - Human Annotation: It is important to note that this step refers to human annotation rather than human collection. The pipeline ensures multi-level filtering, and human experts then annotate the filtered data to confirm their correctness.
>
> 3. Filtering Process and Ratios:
>    The filtering process explains the difference in the ratio of selected bugs to vulnerabilities:
>    - Bug Filtering Pipeline: 40M → 9M → 76K → 3,785 → 248 .
>    - Vulnerability Filtering Pipeline: 14,488 → 249 → 102 → 102 .
>
> 4. The larger difference in the final ratios between  bugs and vulnerabilities arises from the initial dataset sizes: The raw bug dataset starts with 40M samples and the vulnerability dataset starts with only 14,488 samples. It is because vulnerabilities are inherently rarer in real-world data.
>
> We hope this explanation clarifies the collection and annotation process as well as the difference in ratios.
>
>
> >Q:The manual annotation leads to Cohen’s Kappa values like 0.48 and 0.60 which are very low. This means that the annotation quality is rather low.
>
> We believe there has been a misunderstanding of the content in paragraphs 246–250 of our paper. In our paper, we clearly show that we conducted three rounds of annotation, during which the Cohen’s Kappa values progressively improved as follows:
>
>   - **First round:** 0.48
>   - **Second round:** 0.60
>   - **Third round:** 0.88 (final value, reported at line 249).
>
> The final Cohen’s Kappa value of 0.88 demonstrates a high level of agreement and indicates the reliability of our annotations after iterative refinement. This progression highlights our commitment to quality control and the effectiveness of our multi-round annotation process.
>
> We respectfully encourage the reviewer to revisit the relevant section of the paper, as it clearly explains the steps. We hope this clarification addresses your concerns and facilitates a more accurate evaluation of our work.

---

> ### Author Response · Authors · 2024-11-24
>
> > Q:In the annotation process, CWE types were used to help participants decide whether each commit is bug-related. that bugs and vulnerabilities are different concepts. Why use this type of information for bug identification?
>
> Thank you for raising this important point. We agree that bugs and vulnerabilities are distinct concepts, which is why we have treated them as separate ones, with different sources for their collection and analysis (as discussed in our previous response).
>
> We are sorry for any potential confusion in the wording of the paper. To clarify:
>  - CWE types were not used directly to determine whether a commit is bug-related.
>  - Instead, CWE types served as **reference** during the annotation process to help construct a taxonomy of our collected bugs and vulnerabilities. We refer to CWE and build our taxonomy to classify the root causes of both bugs and vulnerabilities systematically. These CWE types were used as an **initial foundation** to guide our classification efforts rather than as definitive criteria for identifying bugs.
>
> We will rephrase the relevant sentences in the revised paper.
>
> >Q: The prompts used for evaluation are simply wrong. Before fixing a bug, we would never know how many code entities we need to modify. That means what we can do is only providing a unified prompt for fixing. cannot ask to perform modifications at a specific level because we do not have this prior knowledge. The experiments conducted in this study were thus totally wrong. and Why can we use prompts of different levels for fixing bugs?
>
> We would like to clarify that in the program repair domain, **fault localization** (identifying the fault lines) and **repair** (generating patches) are typically treated as separate tasks, as established in many state-of-the-art works [1–8]. The standard problem statement for program repair assumes fault localization is provided as input, enabling researchers to focus on the repair process. Our paper follows this widely-used setting, evaluating repair tasks under the assumption that the fault localization is given.
>
> Even under your setting where fault localization is not given, our conclusions remain similar:
>
>  - Despite providing perfect fault localization, the repair performance is still far below expectations, with only a small number of bugs successfully fixed.
>
>  - Hence, if fault localization were not given, we anticipate the results would be similarly poor or even worse, highlighting the inherent challenges of automated program repair.
>
> To further address your suggestion, we conducted an additional experiment to evaluate the fault localization (FL) capabilities of large language models (LLMs). The results, presented in Rebuttal Table 5, indicate that LLMs exhibit low accuracy in identifying fault lines, which negatively impacts the overall repair process. These findings emphasize that separating FL and patch generation allows for targeted improvements in both tasks independently, ultimately enhancing the overall. Specifically, **the lower performance of fault localization significantly affects downstream repair quality. By isolating these tasks, researchers can address their distinct challenges more effectively**.
>
> We will include these discussions in the revised-version. We hope this clarification helps the reviewer better understand the distinction between the two research problems, the commonly-used settings, and the importance of clearly articulating this separation in our work.
>
> RebuttalTable 5. Fault localization performance report on Top@1 of Greedy and TopK=[1,3,5] with temperature 0.8.
>
> |Model||SingleLine|||||MultiLine|||
> |--|--|--|--|--|--|--|--|--|--|
> |  |Top@1[Greedy]|Top@1[T=0.8]|Top@3|Top@5|\||Top@1[Greedy]|Top@1[T=0.8]|Top@3|Top@5|
> |GPT4o-mini|	2.88%|	3.85%|	-|	-	|\||0	|2.76%|	-	|-|
> |CodeLlama-7B-ins|	0	|0|	0.96%	|0.96%|\||	0	|0	|0	|1.69%|
> |DeepSeek-6.7B-ins|	0	|0	|1.92%|	4.81%|\||	0|	0|	6.78%|	8.47%|
> |Grok-beta	|3.85%|	4.81%|	4.81%|	4.81%|\||	0|	0	|3.39%|	5.08%|
>
> \
> \
> \
> References\
> [1] Rahul, Soham, et al. “DeepFix: Fixing common C language errors by deep learning”. AAAI 2017. \
> [2] Sumith, Panupong, et al. “SPoC: Search-based Pseudocode to Code” NeurIPS 2019.\
> [3] Harer, Jacob et al. “Learning to repair software vulnerabilities with generative adversarial networks” NeurIPS 2018\
> [4] Yasunaga, Michihiro et al. “Graph-based, self-supervised program repair from diagnostic feedback” ICML 2020\
> [5] Berabi, Berkay et al. “TFix: Learning to Fix Coding Errors with a Text-to-Text Transformer” ICML 2021\
> [6] Yasunaga, Michihiro et al. “Break-it-fix-it: Unsupervised learning for program repair”,
> ICML 2021\
> [7] Zimin, Steve, et al. “Neural Transfer Learning for Repairing Security Vulnerabilities in C Code” IEEE Transactions on Software Engineering 2022\
> [8] Fu, Michael, et al. “VulRepair: A T5-based Automated Software Vulnerability Repair”  ESEC/FSE 2022

---

> > ### Comment · Reviewer_fJ2S · 2024-11-26
> >
> > Thanks for the detailed response. However, IMHO, the significance of this study is still rather limited due to the following points.
> >
> > 1. The scale of this dataset. In 2014, when Defects4J was proposed, it contained 395 bugs. Now while we are going to enter 2025, we still can only propose a dataset with 248 bugs and 102 vulnerabilities. Can we go deeper than that? How can we ensure the representativeness of the dataset with such a small scale?
> >
> > 2. The Perfect Fault Localization strategy, as argued by the authors, is unpractical while being widely used. In practice, we would not assume that developers know all the code entities to be fixed. From this perspective, the evaluation in the paper, in my mind, would not bring significant meanings.
> >
> > 3. Limited technical contribution. As mentioned earlier, this is a dataset paper with little technical contribution. I understand that ICLR encourages dataset paper as well, but IMHO the process to mine bugs/vuls is well established in the community. I thus see little technical improvement in this paper.
> >
> > I would like to increase my score to 3, but unfortunately, it is still below the bar.

---

> > > ### Author Response · Authors · 2024-11-29
> > >
> > > >Q1, The scale of this dataset. In 2014, when Defects4J was proposed, it contained 395 bugs. Now while we are going to enter 2025, we still can only propose a dataset with 248 bugs and 102 vulnerabilities. Can we go deeper than that? How can we ensure the representativeness of the dataset with such a small scale?
> > >
> > > We gently argue with the reviewer’s misarticulate on such a comparison. We would like to clarify several aspects regarding the scale and representativeness of our dataset:\
> > > \
> > > 1, **High-Quality Dataset Design Purpose:**
> > >
> > > Our dataset includes both a high-quality evaluation test set and a fine-tuning dataset comprising 76K bug-related commits that is much larger than Defect4J. These numbers reflect significant effort and a rigorous filtering process.
> > >
> > > Specifically, we extracted 350 bugs from an initial pool of 40 million commits, following multiple quality checks. While it is possible to increase the number of final bugs, doing so would represent the scaling law of annotation labor. Our primary focus is on ensuring the highest quality, as this is critical for meaningful evaluations.
> > >
> > > We believe that the value of a dataset lies not in its sheer size but in its quality and utility. The scale of our initial raw data (40M commits) far exceeds that of Defects4J, emphasizing the substantial effort behind this work. \
> > > \
> > > 2, **Comparisons with Defects4J:**
> > >
> > > We respectfully disagree with the notion that our dataset must surpass Defects4J in size. The two datasets target different programming languages, projects, and use cases, and they undergo distinct quality filtering processes. These differences make direct comparisons inappropriate.
> > >
> > > - It is important to understand the challenges of stabilizing an executable, reproducible C/C++ defect when facing system library version incompatibilities. For example, the CFLAGS flag -L ssl may rely on different versions of OpenSSL, which can determine the defect's status. In contrast, Java, with its JVM-based environment, is exponentially easier to stabilize compared to our C/C++ efforts.
> > >
> > > - We also address the pain points of existing C/C++ datasets, which require each defect to be equipped with a separate Docker configuration. This approach becomes increasingly impractical for large language models (LLMs). To mitigate this, we designed our dataset to consolidate as many defects as possible into a single container configuration, significantly improving practicality and usability.
> > >
> > > Additionally, 395 bugs (Defects4J) should not be viewed as a standard or baseline number. Recent datasets widely used in the community (Please see following RebuttalTable ) often have fewer than 395 bugs, demonstrating that smaller datasets can still provide significant value when designed thoughtfully.
> > >
> > >
> > > | Benchmarks(non C/C++)  |Classeval[6] |Defects4J 1.0 [10] |JavaBug [7] | GitBug-Java [8] |QuixBugs [9]|
> > > |--|--|--|--|--|--|
> > > |Number|100|357|251|199|80|
> > >
> > > | Benchmarks(Popular) |HumanEval[5] |MBPP[13] |SWEBench-lite [11] |LiveCodeBench(Sep-end) [14]
> > > |--|--|--|--|--|
> > > |Number|164|378|300|349|
> > >
> > >
> > >
> > >
> > > | Benchmarks(C/C++) |Bugs-C++[1] |ManyBugs[2] |Prophet [3] |DBGBench [4] |
> > > |--|--|--|--|--|
> > > |Number|209|185|69|27|
> > >
> > >
> > > To our knowledge, there are no official criteria defining the "ideal" size for a dataset as they have unique features. \
> > > \
> > > 3, **Representativeness of the Dataset:**
> > >
> > >  The representativeness of a dataset is determined by its diversity and quality, not just its size. By focusing on rigorous filtering and careful selection, we ensure that our dataset is representative of real-world bugs and vulnerabilities, even at its scale.
> > >
> > >  The 40M raw bug-related commits we collected for analysis highlight the breadth of our efforts and provide a strong foundation for future expansions of the dataset if needed.
> > > \
> > > \
> > > \
> > > In conclusion, our dataset is designed with a focus on quality, diversity, and practical usability. While its scale is competitive with recent benchmarks, we argue that the real measure of its value lies in its ability to drive the first and meaningful advancements in program repair research.

---

> > > ### Author Response · Authors · 2024-11-29
> > >
> > > >Q2, The Perfect Fault Localization strategy, as argued by the authors, is unpractical while being widely used. In practice, we would not assume that developers know all the code entities to be fixed. From this perspective, the evaluation in the paper, in my mind, would not bring significant meanings.
> > >
> > > We would like to clarify the rationale behind our setting and address your question.
> > >
> > > 1, **Research Problem Definition:**
> > >
> > > - We gently appeal to the reviewer to distinguish between research problems and real-world products. As a research paper, our goal is to address a well-defined problem, which, in this case, is the widely accepted problem statement: “Given fault localization, how to effectively repair the bugs.”  This problem only focuses on repair.
> > >
> > > - It is well-known that fault localization and repair are orthogonal research problems. By focusing on the repair phase, we align with the standard approach adopted in numerous state-of-the-art works. This assumption allows researchers to isolate and improve the repair aspect without conflating it with fault localization challenges.
> > >
> > > Our intention is not to propose an end-to-end product but to contribute to the broader research landscape by advancing the repair task within a well-defined setting.
> > >
> > > 2, **Addressing Fault Localization Concerns:**
> > >
> > > To address your concern about fault localization, we have already conducted additional experiments to evaluate LLMs' capabilities in this area. These results, presented in Rebuttal Table 7, provide insights into LLMs’ performance on fault localization, offering a more comprehensive view of the problem space. We respectfully request the reviewer to review these additional results, which were included previously to address your concerns.
> > >
> > > Rebuttal Table 7. Comparative results (PassRate@1%) of LLMs with and without perfect fault localization in the prompt. Without fault localization guidance, nearly all models fail, particularly commercial general-purpose LLMs.
> > >
> > > |LLM           |w/ or w/o Fault Localization Guidance |Pass@1 (Greedy)|Pass@1(T=0.2)|Pass@1(T=0.8)|
> > > |-----------------|--------------------------------|------|----------|----------|
> > > |CodeLlama-7b-inst|w/o guidance(as reviewer's points)|0     |0.43     |0         |
> > > |                 |w/                         |5.71 |4.49     |4.49     |
> > > |Deepseek-6.7-inst|w/o                   |0     |0.43     |0         |
> > > |                 |w/                         |6.53 |6.53     |6.12     |
> > > |Grok-beta        |w/o                     |0     |0         |0         |
> > > |                 |w/                          |5.71 |6.12     |4.49     |
> > > |GPT-4o-mini      |w/o                    |0     |0         |0         |
> > > |                 |w/                         |6.12 |2.86     |4.90     |
> > > |GPT-3.5-Turbo    |w/o                  |0.86 |0         |0.43     |
> > > |                 |w/                         |7.35 |8.57     |6.94     |
> > >
> > >
> > > 3, **End-to-End Repair Results:**
> > >
> > > To directly address your question, we also added results for end-to-end repair, where no fault localization information is provided. As expected, the results show that none of the bugs could be repaired, further validating our claims that LLMs are currently ineffective for this challenging task without precise fault localization.

---

> > > ### Author Response · Authors · 2024-11-29
> > >
> > > > Q3, Limited technical contribution. As mentioned earlier, this is a dataset paper with little technical contribution. I understand that ICLR encourages dataset paper as well, but IMHO the process to mine bugs/vuls is well established in the community. I thus see little technical improvement in this paper.
> > >
> > > We strongly disagree with the assertion regarding the lack of technical contribution in our paper.
> > >
> > >
> > > 1, **Purpose of a Dataset Paper:**
> > >
> > > Dataset papers serve a fundamentally different purpose than technical papers that propose new methodologies or algorithms. Evaluating a dataset paper based on "technical improvement" is not appropriate, as its primary goal is to provide a valuable resource for the research community. ICLR explicitly encourages dataset papers, recognizing their unique contributions to advancing the field.
> > >
> > > 2, **Limitations of Existing C/C++ Datasets:**
> > >
> > > To the best of our knowledge, the community does not have a well-established, executable process for mining a C/C++ benchmark easily. We have found no existing work or reference supporting the reviewer’s claim that such benchmarks can be mined from a readily available pipeline. Even the closest work, BugCPP[1], has maintained and accumulated its dataset for over two years. Mining executable and reproducible defects for C/C++ is inherently challenging, and most prior datasets do not contribute significantly in addressing these challenges.
> > >
> > > Contrary to the claim that C/C++ datasets are well-established, our work highlights significant limitations in existing datasets, such as:
> > >
> > > - Lack of Diversity: Current datasets fail to capture the breadth of real-world defects.
> > > - Limited Size: Existing datasets are often too small to support robust model evaluation.
> > > - Poor Usability: Many datasets lack reproducibility, making them difficult to use for empirical research.
> > > Our dataset, Defects4C, addresses these limitations by providing a comprehensive, diverse, and reproducible benchmark for C/C++ defects. It sets a new standard for quality and usability in this domain, offering a valuable resource that was previously unavailable.
> > >
> > >
> > > In summary, we believe the reviewer may have misunderstood the purpose and scope of a dataset paper. Dataset research aims to advance the field by providing a foundation for future studies rather than directly proposing new methodologies.
> > >
> > > We also note that some personal opinions expressed in the review may reflect bias. We respectfully express concern about this and encourage a reconsideration of our work based on its stated goals and contributions.
> > >
> > > We firmly believe our paper makes a significant and meaningful contribution to the community by addressing a critical gap in C/C++ defect benchmarking. We respectfully request that our work be evaluated in the appropriate context of dataset research.
> > >
> > > \
> > > \
> > > \
> > > \
> > > Reference\
> > > [1]  Gabin, Minhyuk, et al. "Bugsc++: A highly usable real world defect benchmark for c/c++" ASE, 2023.\
> > > [2] Claire, Neal, et al. "The manybugs and introclass benchmarks for automated repair of c programs." IEEE Transactions on Software Engineering, 2015.\
> > > [3] Fan, Martin. "Automatic patch generation by learning correct code. In Proceedings of " SIGPLAN 2016.\
> > > [4] Marcel, Ezekiel, et al. "Where is the bug and how is it fixed? an experiment with practitioners." FSE 2017.\
> > > [5] Chen, Mark, et al. "Evaluating large language models trained on code."// arXiv preprint arXiv:2107.03374 (2021).\
> > > [6] Du, Xueying, et al. "Classeval: A manually-crafted benchmark for evaluating llms on class-level code generation." ICSE 2024 \
> > > [7] Madeiral F, Urli S, et al. Bears: An extensible java bug benchmark for automatic program repair studies[C] SANER 2019.\
> > > [8] Silva A, et al. GitBug-Java: A Reproducible Benchmark of Recent Java Bugs[C] MST 2024\
> > > [9] Lin D, Koppel J, et al. “QuixBugs: A multi-lingual program repair benchmark set based on the Quixey Challenge[C]”  SIGPLAN 2017 \
> > > [10] Just R, Jalali D, et al. “Defects4J: A database of existing faults to enable controlled testing studies for Java programs[C]” ISSTA 2014\
> > > [11] Carlos, John, et al. "SWE-bench: Can Language Models Resolve Real-world Github Issues?" ICLR 2024\
> > > [12] Mark, Jerry, et al. "Evaluating Large Language Models Trained on Code" arXiv/2107.03374\
> > > [13] Liu, Xia, et al. "Is Your Code Generated by Chat{GPT} Really Correct? Rigorous Evaluation of Large Language Models for Code Generation" NeurIPS 2023. \
> > > [14] Jain, Han, et al. "Livecodebench: Holistic and contamination free evaluation of large language models for code."  arXiv:2403.07974.

---

> ### Author Response · Authors · 2024-12-02
>
> Dear Reviewer,
>
> As the deadline for the discussion period approaches, we kindly request your feedback on our response. Your insights would be greatly appreciated.
>
> Thank you for your time and consideration!

---

### Official Review · Reviewer_VhMf · 2024-10-29

**Soundness:** 3
**Presentation:** 3
**Contribution:** 2
**Rating:** 5
**Confidence:** 3

**Summary:**

This paper introduces Defects4C, a new benchmark dataset for C/C++ program repair, derived from a substantial collection of GitHub commits (9M). The benchmarks consists of 350 defects in total. The authors detail a careful filtering and human verification process. The focus on real-world projects enhances its applicability for assessing program repair methods in practical scenarios. The authors also evaluate the effectiveness of LLM-based APR (automated program repair) on this benchmark.

**Strengths:**

- A new automated program repair (APR) benchmark for C/C++.
- Extensive experiments with 24 LLMs on the benchmark.

**Weaknesses:**

- The novelty and contributions are limited. In recent years, a lot of APR benchmarks have been proposed, covering a variety of programming languages, types (real-world, programming assignments, competitions), difficulty levels, and so on. Table 1 of the paper shows some existing APR benchmarks. There are some more, such as DebugBench (Tian et al., 2024) , EvalGPTFix (Zhang et al., 2023), FixEval (Haque et al., 2023), QuixBugs (Lin et al., 2017b), etc. Therefore, the contributions of this paper are rather incremental.
- The categories of bugs as listed in Table 2 are too rough. I suggest to categorize bugs using finer-grained taxonomy of bugs, which would help reflect the capability of models more comprehensively.
- Lack of evaluations on sub-tasks: The paper only evaluate models on the entire program repair task, without describing the effectiveness of each step, such as bug localization and patch generation. Particularly, given the challenges in identifying the exact location of faults in C/C++ code, the paper could evaluate of LLMs’ capabilities in bug localization, which is important for real-world program repair.

**Questions:**

- What are the novel contributions of this work?
- The dataset includes a large collection of bug-relevant commits (e.g., 9M in total), 248 high-quality buggy functions and 102 vulnerable functions paired with test cases for reproduction. How do you ensure adequate quality control through the human annotation process?

---

> ### Author Response · Authors · 2024-11-24
>
> >Q: The novelty and contributions are limited. In recent years, a lot of APR benchmarks have been proposed, covering a variety of programming languages, types (real-world, programming assignments, competitions), difficulty levels, and so on. Table 1 of the paper shows some existing APR benchmarks. There are some more, such as DebugBench (Tian et al., 2024) , EvalGPTFix (Zhang et al., 2023), FixEval (Haque et al., 2023), QuixBugs (Lin et al., 2017b), etc. Therefore, the contributions of this paper are rather incremental.
>
> Thank you for your detailed feedback and for pointing out these additional benchmarks.
>
> 1. Please refer to our **general response** for a detailed explanation of the key novelties and contributions of Defects4C. While we acknowledge that as a dataset paper, our work may have limited technical novelty, our primary focus has been on creating a **high-quality, realistic, verifiable, highly usable, and sustainable** dataset for C/C++. These features, in our view, are crucial criteria for a dataset to have a lasting impact on the research community.
>
> 2. Among the benchmarks you mentioned, we found that **DebugBench** [8] (Tian et al., 2024) is the only one that includes C/C++ defects. However, these defects are collected from LeetCode and predominantly focus on programming contest problems, which are simpler and may lead to overfitting with modern LLMs. As shown in our experiments (see **general response**), LLMs like GPT-4 and GPT-3.5 achieve a 90–94% pass rate on DebugBench, highlighting that these datasets might not be able to effectively evaluate the current state of APR methods.
>
>  - The other benchmarks, **EvalGPTFix, FixEval**, and **QuixBugs**, focus on defects in Python or Java. Our work specifically addresses C/C++, a domain with unique characteristics:
>
>     - C/C++ programs are more prone to bugs due to their complexity, low-level memory management, and lack of safety features.
>
>     - The collection, confirmation, taxonomy, and even repair methods for C/C++ differ significantly from those for Python and Java.
>
> We sincerely appreciate your contribution to DebugBench and will ensure it is added to the related work section in our revised paper. We also believe that Defects4C addresses critical gaps by providing a robust, realistic, and challenging dataset for C/C++, which sets it apart from existing benchmarks.
>
> | |DebugBench[8] |EvalGPTFix[6]	 | FixEval [7] | QuixBugs [5] |
> |--|--|--|--|--|
> | Language |Java/Python/C/C++ |Java/Python| Java/Python |Java/Python |
> | Source  |Interview/Contest  Leetcode |Interview/Contest AtCoder  |Interview/Contest AtCoder or Aizu  |Interview/Contest Quixey|
> | Size(C/C++) |233  |N/A  | N/A |N/A|
>
>
>
> >Q: Lack of evaluations on sub-tasks: The paper only evaluate models on the entire program repair task, without describing the effectiveness of each step, such as bug localization and patch generation. Particularly, given the challenges in identifying the exact location of faults in C/C++ code, the paper could evaluate LLMs' capabilities in bug localization, which is important for real-world program repair.
>
> Thank you for your insightful suggestion and for highlighting the importance of evaluating sub-tasks like bug localization. Initially, we did not include fault localization evaluations due to resource limitations. However, we have followed the reviewer's suggestion and conducted additional experiments to assess LLMs’ capabilities in fault localization.
>
> **Experimental Setup**: Fault localization aims to identify suspicious locations in the code (e.g., single or multiple statements). For evaluation, we use the Defects4C patch commit line numbers as the ground truth. We provide the complete function content to the LLMs and prompt them to suggest the  fault locations. To avoid data leakage, we do not include error messages or compiler feedback in the fault localization prompts.
>
> We measure the **Top@1/3/5 accuracy**. For multi-line faults, we merge the Top-K most suspicious lines ranked by the LLM’s probability and then compute the metric scores, following prior works [9,10,11].
>
> The results are presented in RebuttalTable 5. and indicate that FL performance still lower in larger general LLM, like Grok-beta. These findings highlight the potential and limitations of LLMs in fault localization, a critical step for real-world program repair tasks.
>
> We will include these results and discussions in the revised version of the paper.
>
> RebuttalTable 5. Fault localization performance report on Top@1 of Greedy and TopK=[1,3,5] when temperature 0.8.
>
> |Model|	|SingleLine|	|		|	| |MultiLine|	|		|
> |--|--|--|--|--|--|--|--|--|--|
> |  |Top@1[Greedy]|Top@1[T=0.8]|Top@3|Top@5| \| |Top@1[Greedy]|Top@1[T=0.8]|Top@3|Top@5|
> |GPT4o-mini|	2.88%|	3.85%|	-|	-	| \| |0	|2.76%|	-	|-|
> |CodeLlama-7B-ins|	0	|0|	0.96%	|0.96%| \| |	0	|0	|0	|1.69%|
> |DeepSeek-6.7B-ins|	0	|0	|1.92%|	4.81%| \| |	0|	0|	6.78%|	8.47%|
> |Grok-beta	|3.85%|	4.81%|	4.81%|	4.81%| \| |	0|	0	|3.39%|	5.08%|

---

> ### Author Response · Authors · 2024-11-24
>
> >Q: The categories of bugs as listed in Table 2 are too rough. I suggest to categorize bugs using finer-grained taxonomy of bugs, which would help reflect the capability of models more comprehensively.
>
> Thank you for your valuable suggestion.
>
> We appreciate your point about using a finer-grained taxonomy. In fact, we have already expanded our categorization into a **three-level** hierarchy, encompassing **35** categories. Due to space constraints, we only presented the high-level categories in the current version of the paper.
>
> For your reference, the following table shows the detailed fine-grained categories, which have been tagged in the evaluation dataset. These finer categories provide a more comprehensive breakdown of bug types, enabling a deeper understanding of model capabilities. please ref to GitHub repository for fine-grained [12].
>
> We will update the paper to include an explanation of this fine-grained taxonomy and how it is used in the dataset.
>
> >Q:The dataset includes a large collection of bug-relevant commits (e.g., 9M in total), 248 high-quality buggy functions and 102 vulnerable functions paired with test cases for reproduction. How do you ensure adequate quality control through the human annotation process?
>
> Thank you for this insightful question. Ensuring the quality of the final evaluation dataset was a key focus of our efforts, and it is precisely why we retained only a small set of high-quality buggy functions (350) after rigorous filtering and annotation. Below, we describe our quality control mechanisms:
>
> 1. We recruited **4 C/C++ experts** with over five years of C/C++ programming experience. Two of these experts are also researchers specializing in software testing, which enhanced their ability to analyze and annotate cases effectively.
>
> 2. Our annotation process involved multiple rounds to ensure accuracy and consistency. Disputes were resolved through in-depth discussions, focusing on the root causes based on code semantics and commit messages.
>
> 3. Note that each annotation was validated by running the associated test cases on both the buggy and fixed versions of the code. This process allowed us to debug and verify whether a given commit truly represents a bug, ensuring high confidence in the annotations.
>
> 4. As described in the paper, we removed ambiguous cases during the annotation process, including those that were either not clear bugs or too difficult to understand or explain. This step reduced the original set of 3,785 commits to the final high-quality dataset.
>
> 5. To measure consistency and reliability among annotators, we calculated **Cohen’s Kappa**, which reached a score of **0.88**. This is considered a high level of agreement and demonstrates the robustness of our annotation process.
>
> We sincerely thank you for your suggestion and will add these details to the revised version of the paper to provide a more comprehensive explanation of our quality control process.
>
> \
> \
> \
> References:\
> [1] Guo, Yuejun, et al. "An investigation of quality issues in vulnerability detection datasets." In 2023 IEEE European Symposium on Security and Privacy Workshops (EuroS&PW)
>
> [2] Croft, Roland, et al "Data quality for software vulnerability datasets." In 2023 IEEE/ACM 45th International Conference on Software Engineering (ICSE)
>
> [3] Kuehn, Philipp, et al "OVANA: An approach to analyze and improve the information quality of vulnerability databases." In Proceedings of the 16th International Conference on Availability, Reliability and Security
>
> [4] Croft, Roland, et al. "Data preparation for software vulnerability prediction: A systematic literature review." IEEE Transactions on Software Engineering 49, no. 3 (2022): 1044-1063.
>
> [5] Lin, Derrick, et al. “QuixBugs: A multi-lingual program repair benchmark set based on the Quixey Challenge” ACM SPLASH 2017
>
> [6] Zhang, Zhang, et al. “A critical review of large language model on software engineering: An example from chatgpt and automated program repair” arXiv:2310.08879
>
> [7] Haque, Md Mahim Anjum et al. “Fixeval: Execution-based evaluation of program fixes for programming problems” IEEE/ACM International Workshop on Automated Program Repair 2023
>
> [8] Tian, Ye, et al. “Debugbench: Evaluating debugging capability of large language models” arXiv:2401.04621
>
> [9] Xia, Deng, et al. “Agentless: Demystifying llm-based software engineering agents” arXiv:2407.01489
>
> [10] Wu, Li, et al. “Large language models in fault localisation” arXiv:2308.15276
>
> [11] Yang, Le, et al. “Large language models for test-free fault localization”  the 46th IEEE/ACM International Conference on Software Engineering 2024
>
> [12] https://github.com/defects4c/defects4c/blob/master/minor_category.md

---

> ### Author Response · Authors · 2024-12-02
>
> Dear Reviewer,
>
> As the discussion period nears its last moment, we would be grateful for your feedback on our response. Your valuable insights will help us further strengthen our work.
>
> Thank you for your time and consideration.

---

### Official Review · Reviewer_mVaS · 2024-10-30

**Soundness:** 3
**Presentation:** 3
**Contribution:** 3
**Rating:** 6
**Confidence:** 5

**Summary:**

This paper presents Defects4C, a dataset of defects in C/C++ mined from open-source repositories using specific commit messages. Defects4C first collects bug-relevant commits, verifies the repositories and commits still exist, and performs certain filtering (e.g., making sure the commits address a single function). Then, it further splits the collected commits into Defects4C_bgcommit, Defects4C_bug, and Defects4C_vul. The datasets Defects4C_bug and Defects4C_vul are rigorously verified by real-life human experts. To understand the effectiveness of LLMs in Automated Program Repair (APR), the authors perform an inference-only and fine-tuning experiments on 24 widely used LLMs. The experimental results indicate LLMs fall behind in fixing C/C++ bugs compared to Defects4J (Java bugs). The authors further introduce a tool for better interacting and reproducing bug datasets.

**Strengths:**

- Mining software repositories to collect C/C++ bugs
- Human analysis of collected bugs and creating bug taxonomy
- Open-sourcing a tool to better interact with the collected bugs
- Large scale study of the effectiveness of multiple LLMs in APR on collected bugs
- Fine-tuning existing LLMs to understand the effectiveness of Defects4C_bgcommit

**Weaknesses:**

- No comparison with other C/C++ bug datasets in the task of APR.
- Some results in Table 3 does not make sense. For instance, why Magicoder result is all 0s. Is there any analysis to root cause this problem because Magicoder is a good model

**Questions:**

Q1. Given that most people are interested in migrating to Rust (because of memory safety features), and there has been research on code translation between C to Rust (https://arxiv.org/pdf/2404.18852), why do you think a dataset of bugs for C/C++ would be worthwhile?

Q2. Have you performed some more analysis on why a model like Magicoder produces 0.0 as result? Do you think the right prompt is not used when inferencing with Magicoder?

Q3. Why didn't the authors perform APR on existing C/C++ datasets. What if overall LLMs are bad on all existing C/C++ datasets including Defects4C? If that is the case, what is the one thing that makes Defects4C stand out?

---

> ### Author Response · Authors · 2024-11-24
>
> >Q1. Given that most people are interested in migrating to Rust (because of memory safety features), and there has been research on code translation between C to Rust (https://arxiv.org/pdf/2404.18852), why do you think a dataset of bugs for C/C++ would be worthwhile?
>
> Thank you for raising this interesting and relevant question.
>
> We would like to address this concern from several perspectives:
>
> 1. The value of our dataset depends on whether C/C++ remains widely used. As long as these languages are in active use, tools for their support, including automated repair, will be necessary. According to a recent study in 2024 [1], C and C++ are still among the top 5 most popular programming languages, and their combined usage would rank them first overall. This continued popularity is why research in areas such as testing, vulnerability detection, code summarization, debugging, and automated repair for C/C++ projects remains vibrant.
> 2. C has the highest number of reported vulnerabilities among programming languages, accounting for over 50% of all disclosed open-source vulnerabilities since 2019 [2]. This statistic highlights the critical need for tools and datasets that support automated repair in these languages, as they directly impact software security.
> 3. While there is exciting research on translating C/C++ to Rust, this technology is still in its early stages and not yet mature. Fully translating complex C/C++ programs into Rust remains highly challenging. Moreover, due to some performance considerations, and practical deployment issues such as hardware constraints and execution speed, Rust is unsuitable in certain contexts, meaning that many companies will likely continue using C/C++ in the foreseeable future. Even if the migration to Rust accelerates in the future, the transition will not be immediate or universal. C/C++ is deeply embedded in many legacy systems, critical applications, and performance-intensive domains. Until these languages are entirely replaced—a scenario that is far from imminent—our dataset will remain valuable for ongoing research and development.
>
> We believe that C/C++ will continue to be popular and relevant in the coming years, making our dataset a significant contribution to the field.
>
>
>
> >Q2. Have you performed some more analysis on why a model like Magicoder produces 0.0 as result? Do you think the right prompt is not used when inferencing with Magicoder?
>
> We conducted the investigation on the magicoder result, the main reason is that magicoder’s output did not follow the LLM regular response which encloses the code by three backticks, i.e. \```, the preprocessing method to extract code content from LLM’s response is a failure, so the result is zero. While this uniform code extractor is employed in most LLMs, i.e. the preprocessor is fair to every LLMs. So if failure to detect the three backticks\``` and it will be marked  fail in pass rate.
>
> We conduct a re-evaluation on magicode for robustness analysis purposes. Assume the magicoder response on the C/C++ bug did not always include any three backticks\```, we report two different scores: utilize the whole response as code even failure to extract code from backticks, or keep the regular preprocessor as other LLMs to only extract code from three backticks\```.
>
> |Preprocessing| Greedy|K=1(T=0.2)|K=10|K=100|K=1(T=0.8)|K=10|K=100|
> |--|--|--|--|--|--|--|--|
> |Code extractor |0	|0	|0|	0	|0|	0|0|
> |Any response(no Code extractor)|	3.25	|2.63|	9.9	|24.651|	4.72	|22.55| 34.84|
>
>
>
> We will revise the submission with different preprocessor results, and explain the magicoder’s zero scores.
>
> >Q3. Why didn't the authors perform APR on existing C/C++ datasets. What if overall LLMs are bad on all existing C/C++ datasets including Defects4C? If that is the case, what is the one thing that makes Defects4C stand out?
>
> Thank you for the suggestion. Please refer to the general response.
>
> Reference:\
> [1] https://distantjob.com/blog/programming-languages-rank/
>
> [2]https://www.mend.io/most-secure-programming-languages

---

> > ### Comment · Reviewer_mVaS · 2024-11-25
> >
> > Thank you authors for your response. It would be great to perform more deep checks to make sure there are no more backtick problems. I have dealt with this problem myself and writing a proper extractor can be a bit challenging. I would suggest to suffix prompts with ``` so the model could properly fill in the rest of the content. This way you are guaranteed to have backticks to extract the response. Please see below:
> >
> > ````
> > @@ Instruction
> > fix code in line XX.
> >
> > @@ Response
> > ```
> > ````
> >
> > Also, I am glad to raise my score to 6. Thanks.

---

### Official Review · Reviewer_rPB6 · 2024-11-03

**Soundness:** 3
**Presentation:** 2
**Contribution:** 3
**Rating:** 6
**Confidence:** 4

**Summary:**

The paper develops a benchmark for Automated Program Repair (APR) in C/C++ programs. The authors curate a dataset of C/C++ bug-fixing commits, apply systematic filtering, and use human annotations to ensure data quality. The benchmark is evaluated through various experiments using multiple large language models (LLMs), with a comprehensive analysis of the results. The main contribution is providing training set and a benchmark for C/C++ APR task, which can be used for evaluating code LLMs.

**Strengths:**

- The authors introduce a new benchmark for APR with C/C++ with human annotation.
- The benchmark has been well-categorized.
- A training set for APR tasks in C/C++ is available.

**Weaknesses:**

- Human Annotation Process: More details about the human annotation process should be discussed in Section 3.3. Please refer to Questions for suggestions.

- Experimental Results: The authors only represent the performance of LLMs. The performance of traditional APR methods is missing. The authors also miss some of the latest LLMs with the most competitive performance on other benchmarks. Please refer to Questions for suggestions.

- Presentation: Some contents should be relocated to make the paper more organized. The paper contains some issues on grammar and style of writing.

  - The introduction is lengthy; some contents could be relocated to Related Work or Benchmark Construction. For example, paragraphs describing the dataset collection could be made more concise (line 81 to 103).

  - Line 83: Change “vulnerability functions” to “vulnerable functions.”

  - Line 241: Explain CWE upon its first use.

  - Line 383 in Table 3: Correct “GPT-35-Turbo” to “GPT-3.5-Turbo.”

**Questions:**

- Human Annotation Process:
  - In lines 248-249, you mention that half of the bugs were reviewed in the third round. What about the other half? Does this imply that some parts of the dataset received two rounds of annotation, while others received only one? Additionally, is the reported Cohen’s Kappa calculated based on only the reviewed half or all annotated commits?
  - In line 255, it’s noted that 248 commits were identified for Defects4C_bug. This is a significant reduction from the original 3,785 commits. Does this reduction imply that the remaining commits in the 3,785 were deemed non-bug-related, which could affect training set quality, or were only a portion of the 3,785 commits annotated by human experts?

- Experimental Results:
  - Could you include some traditional APR methods as baselines in Table 3? The paper frequently claims that LLMs perform well, and a traditional baseline would strengthen the claim.
  - For close values in Table 6, standard errors are needed to support the claim that fine-tuning improves model performance.
  - Including evaluations of the latest general-purpose LLMs, like GPT-4o(-mini), Claude3, Gemini 1.5, or even o1-preview, would be helpful. Testing one or some of these models could demonstrate how recent models perform on C/C++ APR tasks.
  - Given that Defects4J is an older dataset, it’s possible that knowledge contamination is affecting model performance. Conducting a case study on this potential contamination could provide valuable insights. Furthermore, are there methods to prevent your benchmark from contaminating future LLM knowledge?

---

> ### Author Response · Authors · 2024-11-24
>
> >Q1 In lines 248-249, you mention that half of the bugs were reviewed in the third round. What about the other half? Does this imply that some parts of the dataset received two rounds of annotation, while others received only one? Additionally, is the reported Cohen’s Kappa calculated based on only the reviewed half or all annotated commits?
>
> Thank you for pointing out this confusion in our description.
>
> To clarify, during the third phase of annotation, we aimed to improve annotation quality by splitting the entire dataset into two halves. Initially, we annotated the first half and checked the agreement on these results, conducting discussions as needed. This approach allowed us to ensure that, if the agreement on the first half had been unacceptably low, additional refinement (i.e., the fourth round) would be performed. However, we found that the agreement on the first half was already acceptable, so we proceeded to annotate the second half using the same process.
>
> Importantly, the reported Cohen’s Kappa value was calculated based on the entire dataset, encompassing both the first and second halves.
>
> We will revise the unclear phrase.
>
> >Q2 line 255, the 248 commits is a significant reduction from the original 3,785 commits. Does remaining commits in the 3,785 were deemed non-bug-related, that could affect training set quality, or were only a portion of the 3,785 commits annotated by human experts?
>
> Thank you for your insightful question.
>
> As explained in the paper, we employed a rigorous and conservative human annotation process to ensure the highest quality for the evaluation dataset. This process resulted in the selection of 248 confirmed commits.
>
> As discussed in Lines 251–256, the reduction from 3,785 commits is due to the following:
>
> 1. A substantial portion of the commits were non-bug-related, as verified through human inspection.
>
> 2. Some commits contained bug fixes that were too vague or ambiguous to be definitively classified as real bugs.
>
> To avoid introducing noise or uncertainty into the evaluation dataset, we prioritized quality over quantity and included only the 248 commits that were unequivocally confirmed as bug-related.
>
> We will clarify it more clearly in the paper.
>
> >Q3 Could you include some traditional APR methods as baselines in Table 3? The paper frequently claims that LLMs perform well, and a traditional baseline would strengthen the claim.
>
> Thank you for this thoughtful suggestion.
>
> Before submitting the paper, we evaluated 10 traditional APR methods as potential baselines. Unfortunately, the majority of these methods (e.g., Prophet, Darjeeling, F1X, Verifix) could not be successfully executed for the following reasons:
>
> 1. Many traditional APR tools are designed for small programs with a single function and lack support for building and verifying large-scale projects.
>
> 2. Some tools rely on resource-intensive techniques, such as symbolic execution, which are not scalable for large projects.
> Ultimately, we were able to successfully execute two traditional tools, **VRepair** and **VulRepair**. As shown in RebuttalTable 2, these tools performed significantly worse compared to LLM-based methods, successfully repairing only one bug.
>
> In fact, recent advancements in APR research have consistently demonstrated the superior performance of LLM-based methods, which are now regarded as the mainstream approach [2]. This rationale underpins our decision to focus on multiple LLMs for the evaluation.
>
> RebuttalTable 2. New results of non-LLM baselines( original Tab.4)
>
> |Model|Decoding|Function| |Hunk| |Line| ||
> |--|--|--|--|--|--|--|--|--|
> |---| 	----|#P/T| #Avg.t| #P/T| #Avg.t|	#P/T| #Avg.t|	Sum |
> |CodeLLama-7B-ins|	T=1	|0/17	|0|	18/19|	2.14	|7/69|	3.55	| 25|
> |VulRepair[1]|	Beam Search	|0/17	|0| 0/19 |   0    |1/69|	10    | 1|
> |VRepair[2]|	Beam Search	  | 0/17 |	0|	0/19	|0	|0/69	|0	| 0 |
>
>
> We appreciate your feedback and will incorporate the results of non-LLM methods, along with detailed discussions, in the revised version to further strengthen our claims.
>
>
>
> >Q4: For close values in Table 6, standard errors are needed to support the claim that fine-tuning improves model performance.
>
> Thank you for your constructive suggestion.
>
> In response, we have included the standard errors in RebuttalTable 3, calculated based on results from 5 random seeds. The results indicate that while fine-tuning shows some improvements in certain cases, it does not consistently enhance performance across all metrics. We appreciate your feedback and will ensure that the updated table and discussion are included in the revised version of the paper.
>
> RebuttalTable 3. Comparative Results of LLMs With Fine-Tuning (original Tab 6.).
>
> |Model|Greedy|T=0.2(k=1)|T=0.8(k=1)|
> |--|--|--|--|
> |Codellama-7B-Base|0.08±0.18|0.29±0.18|0.25±0.23|
> |Codellama-7B-Instruct|3.87±0.42|4.00±0.40|3.73±1.11|
> |Deepseek-Coder-6.7B-Base|0.43±0.05|0.22±0.23|0.45±0.13|
> |Deepseek-Coder-6.7B-Instruct|	6.71±2.17|	6.35±1.65|5.61±1.39|

---

> ### Author Response · Authors · 2024-11-24
>
> >Q5: Including evaluations of the latest general-purpose LLMs, like GPT-4o(-mini), Claude3, Gemini 1.5, or even o1-preview, would be helpful. Testing one or some of these models could demonstrate how recent models perform on C/C++ APR tasks.
>
> Thank you for your valuable suggestion.
>
> In response, we have conducted additional evaluations on some of the latest general-purpose LLMs, specifically GPT-4o-mini, o1-preview, and Grok. As shown in RebuttalTable 4, while these recent models exhibit promising performance compared to existing LLMs like DeepSeek and CodeLlama, they still face significant challenges in addressing C/C++ APR tasks.
>
> RebuttalTable 4 More latest general-purpose LLMs. (original Tab4.)
>
> |Model|Greedy|T=0.2(k=1)|T=0.8(k=1)|
> |--|--|--|--|
> |GPT-4o-mini|6.12| 2.86| 4.90|
> |Grok-beta|5.71|6.12|4.49|
> |GPT-o1-preview|3.4|4.2|4.5|
> |GPT-3.5-Turbo|7.35|8.57|6.94|
>
> We appreciate your recommendation and will include the results and detailed discussions in the revised version of the paper to provide a more comprehensive evaluation of these models.
>
> >Q6:Given that Defects4J is an older dataset, it’s possible that knowledge contamination is affecting model performance. Conducting a case study on this potential contamination could provide valuable insights. Furthermore, are there methods to prevent your benchmark from contaminating future LLM knowledge?
>
> We appreciate the reviewer’s insightful comments regarding the potential contamination impact on our dataset.
>
> Indeed, we acknowledge that there is knowledge contamination in the case of **Defects4J** used in our results. For example, we observed that some models could still repair buggy functions even when we removed necessary variables or definitions. We will incorporate these findings and their implications in the paper.
>
> Regarding our own dataset, we have implemented several measures to mitigate contamination risks:
>
> 1. We ensured that the patches include corresponding timestamps, allowing us to enforce a knowledge cut-off date. This enables users to evaluate the temporal difference between the dataset and the LLM training data.
> 2. Our dataset construction process is fully automated for collecting new bugs (after the training of LLMs), with only the human annotation phase requiring manual effort.
> 3. We implemented a decontamination script based on Magicoder’s algorithm, which identifies and removes samples containing high-similarity substrings from the training corpus, thereby reducing overlap.
> 4. To further mitigate contamination risks, we are developing a mutation-based approach to semantically preserve the dataset while introducing changes such as variable renaming, stylistic adjustments, and code sequence modifications. This will ensure that the bugs remain unaffected while making the dataset less susceptible to contamination.
>
> We appreciate the reviewer’s suggestion and will include these measures and discussions in the revised paper to address contamination concerns comprehensively.
>
> \
> \
> \
> References:\
> [1] Yasunaga, M., Liang, P.. Graph-based, self-supervised program repair from diagnostic feedback. In ICML’20.
>
> [2] Xia, C. S., Wei, Y., & Zhang, L. Automated program repair in the era of large pre-trained language models. ICSE’23

---

> > ### Comment · Reviewer_rPB6 · 2024-11-26
> > **Response to the Authors' Rebuttal**
> >
> > I appreciate the authors’ efforts in addressing most of my concerns, especially about the contamination part.
> >
> > One issue regarding fine-tuning remains partially-solved. In Rebuttal Table 3, I am expecting a comparison between the fine-tuned model and the original model, which necessitates including the standard errors for both. The current table only provides standard errors for the fine-tuned models.
> >
> > Additionally, you stated that the standard errors are based on 5 random seeds—did you use bootstrapping for this calculation?
> >
> > Please clarify and include these details in the paper, possibly in the appendix. Moreover, if the goal is to demonstrate the improvement of fine-tuned models on certain metrics, calculating the p-value to test whether the distributions differ could provide stronger statistical evidence if the mean and standard error are not obvious proof.
> >
> > Another concern is still the number of identified bugs and vulnerabilities from only 248 confirmed commits. The number may not be large and comprehensive enough to cover different types of defects in C/C++ programs, given their prevalence.
> >
> > I would like to keep the current rating.

---

> > > ### Author Response · Authors · 2024-12-03
> > >
> > > >Q7 One issue regarding fine-tuning remains partially solved. In Rebuttal Table 3, I am expecting a comparison between the fine-tuned model and the original model, which necessitates including the standard errors for both. The current table only provides standard errors for the fine-tuned models.
> > >
> > > We sincerely thank you for your valuable comments, which are professional, rigorous, scientific, and highly helpful in refining our paper.
> > >
> > > In Rebuttal Table 3, and response to your suggestion, we have now included the standard errors and p-values for both models. These updated results can be found in the [appendix (page 21)](https://github.com/defects4c/defects4c/blob/master/Defects4C_paper.pdf), highlighted in blue. Please note that our results do not include bootstrapping.
> > >
> > >
> > >
> > > >Q8 Another concern is still the number of identified bugs and vulnerabilities from only 248 confirmed commits. The number may not be large and comprehensive enough to cover different types of defects in C/C++ programs, given their prevalence.
> > >
> > > Our dataset size aligns with the conventions of current benchmarks, such as HumanEval (164), MBPP (378), and SWEBench-lite (350), as shown in Table 15 in the appendix. \
> > > While our final results involve 350 bugs, these were derived from many projects and commits. This reduction reflects our focus on **data quality, scope, and the requirement for unit tests** (more detail in appendix page 23):
> > >
> > > - Data Quality: Recent studies  emphasize the low quality of many existing bug/vulnerability databases. We implemented strict filtering criteria to ensure the genuine, reproducible, and high-quality nature of our bugs.
> > > - Scope Limitation: To prioritize quality and verifiability, we focused on bugs within single functions, simplifying human confirmation and reproduction.
> > > - Unit Test Requirement: We required unit tests for all bugs, excluding those without corresponding tests.
> > >
> > > These stringent criteria reduced the number of viable commits from 38 million to 76,000. However, we believe our dataset size remains comparable to existing benchmarks, particularly in terms of defect types that you may concern. Because, given the large and diverse initial pool, which offers a wide range of defects, combined with expert human annotation that selectively chooses representative samples,  and the automatic process is highly reliable and capable of expanding in future versions. For instance, widely used datasets such as Defects4J 1.0 contain only 357 defects, which is similar in scale to our dataset. As the first version of Defects4C, we collected 350 bugs and are committed to expanding our dataset in subsequent versions.
> > >
> > > Reference:\
> > > [1] more rebuttal updated in Revised paper in https://github.com/defects4c/defects4c/blob/master/Defects4C_paper.pdf

---

### Author Response · Authors · 2024-11-24
**General Response**

We thank all reviewers for their valuable comments. While three of the reviewers have recognized the contributions of our paper, including the introduction of a valuable dataset from real-world projects, detailed human analysis and bug taxonomy, open-source tools, and a large-scale study, we also note that Reviewer fj2S has significant misunderstandings and biases regarding the paper.

The major concerns raised by Reviewer mVaS and Reviewer rPB6 focus on how our dataset, Defects4C, differs from existing datasets. Below, we provide a common response to highlight the contributions of our dataset. While individual responses are also provided to address specific concerns separately.

**Contribution of Defect4C**

The primary motivation for constructing Defects4C is to address key limitations of existing C/C++ datasets, as discussed in our paper:

1. Recent **research [4-9]** has shown that existing bug/vulnerability databases often suffer from low quality, making them challenging to use in bug-related research.
2. Specifically, many existing datasets, such as DebugBench [2] and Codeflaw [1,3], are based on student assignments, interview questions, or contest problems. While useful, these datasets fail to reflect the complexity of real-world programs.
3. Some datasets are derived from real-world projects, but their small size and limited diversity make them insufficient for robust evaluation and even fine-tuning.
4. Many existing datasets lack proper engineering support, making them challenging to run and integrate into modern repair frameworks. This usability issue is critical for practical research.


RebuttalTable.1  Existing datasets fall short in capturing the complexity of real-world programs. The strong performance(pass@1 Greedy) of LLMs on these datasets indicates that they are relatively simplistic and fail to adequately challenge or assess the effectiveness of current APR methods in real-world scenarios.

| Benchmark in C/C++ | Source | GPT-3.5-Turbo | GPT-4 | Codellama-34-inst |
| ---------- | ------------ | ------------ | ------------------ | ----------- |
| DebugBench  [2]     | Interview/Contest (Leetcode)          | 59.0% | 74.6% | 16.4 %         |
| Codeflaw  [1,3]     | Interview/Contest (Codeforces)          | 94.0% | 93.0% | 91.0%         |
| Defects4C(our)     | Real-World          | 9.0% | 8.5% | 4.0%         |


Following the suggestion of Reviewer mVaS, we evaluated recent LLMs on existing datasets. As shown in RebuttalTable 1., LLMs perform well on these datasets, achieving high scores. This result suggests that existing datasets are **relatively easy and predominantly comprise simple programs**, which do not sufficiently evaluate current APR methods in real-world programs.

We made substantial efforts to construct Defects4C as a high-quality dataset for the community, inspired by the mature design of Defects4J. The contributions of our dataset include:

1. **Broader range of bug types:** Defects4C covers significantly more samples and projects than existing real-world datasets, providing data for both training and rigorous evaluation.
2. **Ease of use:** Our framework includes a well-designed framework and scripts for reproducing and validating bugs, making Defects4C easy to integrate with various repair methods.
3. **Comprehensive evaluation:** We conducted a large-scale evaluation of LLM-based repairs, revealing limitations in current methods and emphasizing the potential of our dataset to drive future improvements.
4. **Mitigation of data leakage and contamination:**
   * Patches include corresponding timestamps, allowing users to enforce a knowledge cut-off based on the LLM training date.
   * Our dataset construction is automated, enabling the easy collection of new bugs for testing future LLMs while ensuring temporal separation.
   * Following Magicoder’s algorithm, we implemented a decontamination script to remove overlapping samples with high-similarity substrings.
   * We are extending our dataset by applying semantic-preserving transformations, such as variable renaming, stylistic changes, and code sequence modifications, to enhance robustness.

In summary, Defects4C stands out as a challenging and realistic dataset designed to push the boundaries of current APR techniques. By addressing critical gaps in existing datasets, we believe that Defects4C will significantly benefit the research community and contribute to advancing automated program repair methods.

---

### Meta-Review · Area_Chair_s3xh · 2024-12-21

**Metareview:**

The paper develops a benchmark for Automated Program Repair (APR) in C/C++ programs. The authors curate a dataset of bug-fixing commits, apply systematic filtering, and use human annotations to ensure data quality. Inference-only and fine-tuning experiments conducted on several LLMs provide insights into their ability to handle coding tasks. The main issue is the limited contribution: while the paper introduces a new benchmark for C/C++, it lacks a fine-grained evaluation and is not comprehensive enough for the C/C++ language. Additionally, there are many aspects that need to be added and revised. As a result, it is a standard contribution but requires further improvement.

**Additional Comments On Reviewer Discussion:**

Key Discussions:
- Clarifications:
  - The authors clarify that the main contribution is creating a high-quality, realistic, verifiable, highly usable, and sustainable dataset for C/C++.
  - Description of the dataset collection and human annotation process.
  - Some experimental analysis, including table results and setup details.
- Additional Experiments:
  - Comparison with traditional APR methods and more LLMs.
  - Evaluation of LLMs’ capabilities in fault localization.

The main concerns focus on the limited contribution and the lack of insights from the experiments. In my opinion, the primary contribution is a high-quality benchmark. However, for a benchmark, it lacks a deeper exploration of the C/C++ programming process and a fine-grained evaluation.

---

### Decision · Program_Chairs · 2025-01-22

Reject

---

> ### Public Comment · ~Jian_Jornbowrl_Wang1 · 2025-10-15
> **Directing Readers to the Latest ASE-Published Version of Defects4C**
>
> Dear readers
>
> Defects4C is live on arXiv https://arxiv.org/abs/2510.11059. Try the C/C++ benchmarking suite for your LLM evaluation and share feedback. Thanks to the reviewers and program chairs for their support.
>
> Best regards
>
> authors